# Anterior cingulate neurons combine outcome monitoring of past decisions with ongoing movement signals

Lukas T. Oesch ⬦ , Makenna C. Thomas, Davis Sandberg ⬦ , João Couto ⬦ & Anne K. Churchland ⬦ ✉

In dynamic environments, animals must closely monitor the effects of their actions to inform switches in behavioral strategy. Anterior cingulate cortex (ACC) neurons track decision outcomes in these environments. Yet, it remains unclear whether ACC neurons similarly monitor behavioral history in static environments and, if so, whether these signals are distinct from movement representations. We recorded large-scale ACC activity in freely moving mice making visual evidence-accumulation decisions. Many ACC neurons exhibited nonlinear mixed selectivity for previous choices and outcomes (trial history) and were modulated by movements. Trial history could be stably decoded from population activity and accounted for a separable component of neural activity than posture and movements. Trial history encoding was conserved across different subjects and was unaffected by fluctuating behavioral biases. These findings demonstrate that trial history monitoring in ACC is implemented in a conserved population code that is independent of the volatility of subjects' task environment.

A key problem for decision-makers is to discover a set of rules or internal models that map cues in the environment to actions in a way that yields a desired outcome. To build or update internal models, animals must explore different response options and their resulting outcomes and then integrate over these past experiences to determine the best course of action. The anterior cingulate cortex (ACC) is believed to play a major role in evaluating the outcomes of ongoing behavior to inform changes in behavioral strategy[1]. Evidence supporting this idea comes from several lines of research in humans, non-human primates, and rodents performing probabilistically rewarded tasks in dynamic environments where the rules governing which actions are rewarded frequently switch. In these tasks, ACC neurons encode predictions about expected rewards resulting from a specific behavior[2,3]. Violations of these predictions (that are sometimes called surprise signals or unsigned prediction errors) strongly drive neural activity in the ACC[4,5]. One hypothesis is that these prediction error signals are then used to update subjects' current beliefs and the exploration of alternative behaviors[6–13]. In support of this hypothesis, inhibition of ACC neurons prevent animals from adapting their behavior to a new rule after changes of the task environment[12–15]. This view of ACC function would predict that the post-outcome activity of ACC neurons should generally be low and would only be elevated after an unexpected outcome, signaling a change in the environment[7]. Surprisingly, ACC neurons not only strongly encode previous choices and outcomes after block transitions when the subjects' behavior needs to be updated, but also within blocks when subjects exploit their task knowledge[16–18].

This existing work suggests that the ACC might not only drive belief updating but also continuously monitor trial history. However, most studies investigating the functions of ACC have employed tasks where decision-rules switch in a block-wise manner. In these tasks, close monitoring of trial history and tracking of unexpected changes is built into the task design and is required for high task performance. A new approach is needed for two reasons. First, imposing a block

Department of Neurobiology, David Geffen School of Medicine, University of California, Los Angeles, Los Angeles, CA, USA.
✉e-mail: AChurchland@mednet.ucla.edu

structure on behavioral tasks, as is done in dynamic environments, can lead to strong correlations between previous choices and outcomes, as well as previous and upcoming reporting actions[16]. It is thus unclear if the recorded signals in ACC neurons reflect true trial history monitoring or, instead, postural or movement variables[19]. Second, it remains unknown whether trial history signals in ACC neurons exclusively emerge in dynamic, unpredictable environments or whether they may also be present in deterministic tasks with stable rules governed by observable sensory inputs. If trial history signals are present even in these static environments, this would argue that decision circuits are wired with flexibility in mind, allowing decision-makers to constantly evaluate their strategy. Previous behavioral observations hint at this idea: perceptual decisions in stable environments show hallmarks of strategies for economic decisions in dynamic environments[20]. These strategies, which include uncertainty-guided exploration[21], rely on retaining information about outcomes of previous decisions. The ACC is a natural candidate to track such choice–outcome combinations, but its responses have been studied almost entirely in the context of dynamic, economic decisions.

To overcome these limitations, we trained mice in a freely moving perceptual evidence accumulation task with a fixed rule and a randomized trial structure[22,23] while recording neural activity from ACC neurons and carefully tracking animals' posture (position in space and head-orientation angles) and movements. In keeping with the hypothesis that ACC monitors trial history, we found that ACC neurons strongly encoded choice- and outcome history and their different interactions. Using decoding analyses, we show that these representations remain stable well into the new trial. Taking advantage of methods to separate the influence of often highly collinear task variables and movements on neural activity[24,25], we show that trial history signals cannot simply be explained by animals' body position or movements. Finally, we demonstrate that trial history, but not movement encoding, is low-dimensional and that the neural dynamics encoding trial history are conserved across subjects.

## Results

### Perceptual decisions in expert performers are modulated by trial history

We trained mice on a freely moving visual decision-making task that required the integration of pulsatile sensory evidence over time[22,23]. After initiating a trial by poking into the center port, mice were presented with a train of visual flashes (Fig. 1a, b). On trials where the number of flashes exceeded the category boundary of 12 events per second (12 Hz), mice were rewarded for poking into the right port and conversely, were rewarded for choosing the left port after a low-rate stimulus trial. Importantly, these contingencies remained stable and were not reversed. Furthermore, whether a trial was high- or low-rate was randomly assigned. These measures ensured that the environment remained static and that there was no trial structure that animals could exploit by using trial history information. Animals learned this task (center fixation and stimulus discrimination) over the course of $10.72 \pm 1.25$ weeks and achieved high performance levels (Fig. 1c).

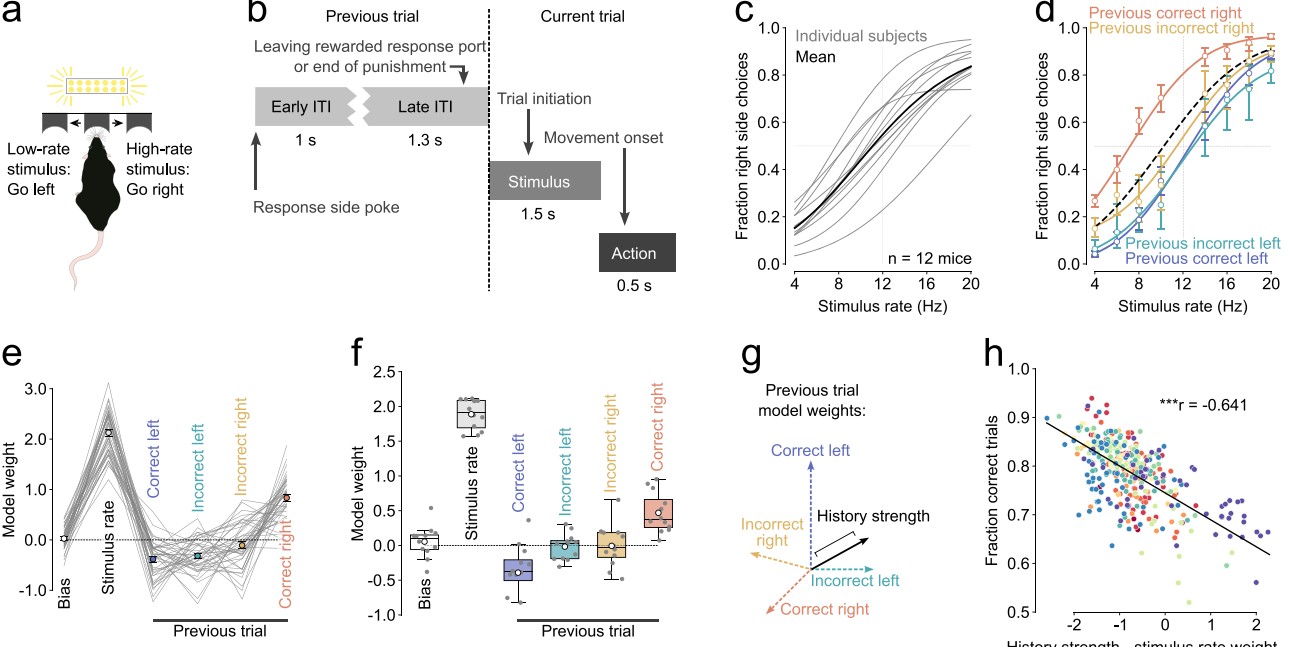

**Fig. 1 | Perceptual decisions in expert performers are influenced by trial history. a** Illustration of perceptual task setup. Animals initiate trials by poking into a central port. This triggers a 1 s train of visual flashes. Animals are required to wait inside the center port for one second before reporting their decisions in either the left or the right port. Animals are rewarded for choosing the left port after low-rate stimuli and for choosing the right port after high-rate stimuli. Incorrect choices result in a 16 kHz tone and 2 s timeout. **b** We divided trials into four distinct phases: Early ITI: the 1 s period following the animal's choice report, Late ITI: 1 s before to 0.3 s after leaving the chosen port after correct choices or after the end of the punishment sound and timeout after incorrect choices, Stimulus: 0.5 s before to 1 s after stimulus onset, Action: 0.2 s before to 0.3 s after reporting movement onset. **c** Psychometric functions for individual subjects (grey, $n = 12$ animals, 356 sessions) and averaged across subjects (black). **d** Psychometric functions for trials that were preceded by specific combinations of previous choice and outcome (trial history) for an example subject. Dashed black line represents the overall fit, scatter points and error bars are the mean ± Wilson score interval ($n = 13,334$ trials from 39 sessions). **e** Weights of a logistic regression trained to predict choice based on the stimulus rate and trial history from the subject shown in (**d**). Lines represent individual sessions, dots show mean ± sem weight across sessions ($n = 38$ sessions). **f** Boxplot showing the median ± interquartile range of the model weights across subjects. Whiskers depict 1.5 × the interquartile range. White dots represent the mean and dark scatter points are individual subjects ($n = 12$ animals). **g** Behavioral influence of trial history quantified as the length of the vector in the space spanned by the four distinct combinations of previous choice and outcome. **h** The difference between history strength and stimulus rate weight is negatively correlated with task performance. Points depcit individual sessions and are color-coded by subject identity. $P = 1.62 \cdot 10^{-42}$, two-sided permutation test for Pearson's correlation coefficient.

A series of recent studies have suggested that perceptual decisions might be influenced by previous choices and/or outcomes even in expert performers[23,26–31] and that previous correct and incorrect choices might differentially impact subsequent behavior[32]. To verify that mice in our study were similarly influenced by trial history, we partitioned subjects' trials from all the sessions based on whether the previous choice was to the left or right and whether the mice were rewarded for their previous choice. We then fitted a psychometric function to trials with these different trial history contexts and found that the curves were indeed different. Figure 1d shows the psychometric curves for an example mouse whose choices became right biased after correct right choices but were largely unaffected by previous incorrect right choices and previous left choices regardless of their outcome (Fig. 1d). To test which aspects of behavior were particularly influenced by trial history, we obtained estimates for the perceptual bias, sensitivity, upper- and lower lapse rates for all the mice by fitting psychometric curves to their session performance. We found that trial history only significantly changed the subjects' perceptual bias (Supplementary Fig. 1a–d). To compare how strongly animals' decisions were influenced by sensory evidence or specific trial history contexts on a session-by-session basis, we modeled the animals' choices as a function of the current stimulus rate plus an individual regressor for each trial history context (Fig. 1e). As expected, we found that for all the expert mice, the model weights for stimulus rate strongly influenced animals' decisions (Fig. 1f). Trial history biased animals' choice in different ways. While animals tended to repeat previously rewarded choices (negative signs indicate favoring left choices), previous incorrect choices had a weaker and less consistent influence on upcoming decisions (Fig. 1f). Nevertheless, the magnitudes of all the model weights were significantly larger than the weight magnitudes from a shuffled control (linear mixed-effects model with subject as random effect). The weaker average influence of incorrect choices was not because they had large values that could be either positive or negative. Indeed, the session-by-session variance of all the trial history weights was similar (Supplementary Fig. 1e) and we found no evidence of clusters of sessions with similar weight patterns (Supplementary Fig. 1f). Importantly, when we looked not one- but two trials back, we found that the choices, outcomes and their interaction had a lower impact on the mice's decisions than the ones from the last trial and they were very close to zero (Supplementary Fig. 1g).

Given that the trial structure of our task was random and that only the stimulus rate determines the rewarded side, using trial history to inform choices will decrease performance. To verify this, we first summarized the overall influence of the most recent trial history on the animals' choices (history strength) by taking the length of the vector of all trial history weights (Fig. 1g). We then subtracted the stimulus weight from the history strength and correlated this difference to the animals' performance. We found a strong negative correlation, indicating that performance decreases as history strength grows (Fig. 1h). These findings confirm that decisions in expert mice are influenced by multiple trial history contexts even when this leads to suboptimal performance[33].

## Trial history can be decoded from ACC population activity

We then asked whether ACC neurons monitor combinations of choices and outcomes and whether these cells maintain this information as a trial history signal into the new trial in expert performers (Fig. 1b). To this end, we recorded neural activity from CaMKII-expressing neurons in the ACC using miniaturized head-mounted fluorescence microscopes[34] (Fig. 2a, b). The small, lightweight size of these scopes meant that we were able to measure neural activity in freely moving mice. Further, this approach allowed us to record many neurons simultaneously ($322 \pm 42$ neurons per subject) and repeatedly sample neural activity from the same subjects over multiple sessions, enabling the use of powerful population level analyses and comparisons within- and across subjects.

The activity of many ACC neurons was modulated by specific trial history contexts (Fig. 2c). For example, the activity of some neurons was driven almost exclusively by previous correct right or correct left choices during the inter-trial interval (ITI, Fig. 2d, N1, N4) or by both previous incorrect left and right choices but at different times within the ITI (Fig. 2d, N7), indicating clear non-linear mixed selectivity[35,36]. To summarize the selectivity of the population of all ACC neurons we plotted the activity difference between previous right and left choices against the difference of activity for previous correct and incorrect trials (Fig. 2e). We found a large variety of different response profiles. Notably however, while the population tended to show higher activity for previous incorrect trials than for correct ones, neurons were more selective for previous right or left choices after correct than after incorrect trials.

To quantitatively assess the strength of neural representations of trial history, we trained linear decoders on ACC population activity to predict the different trial history contexts (4 classes). The decoding accuracy over the entire trial time was high (78% as compared to 25% at chance level) and significantly higher than the shuffled control (Fig. 2f). Depending on the structure of neural representation, optimizing decoders to classify all the non-linear combinations of previous choice and outcome might lead to losses in decoding accuracy of the underlying binary decoding of previous choice or previous outcome (Supplementary Fig. 2a). However, we found no difference in the classification accuracy for previous choice or previous outcome between the full trial history decoders and binary classifiers specifically optimized to decode previous choice or outcome (Supplementary Fig. 2b). This suggests that neural representations in the ACC support independent readouts of previous choice and outcome and their combinations. Next, we asked whether all trial history contexts could be decoded equally well during all phases of the trial. To answer this question, we separately computed the decoding accuracy for each trial history context across the different trial phases (Fig. 2g). We found that the temporal dynamics depended on the outcome of the previous trial: the decoding accuracy of previously rewarded left and right trials was similar and remained high throughout the ITI phases and slowly decayed thereafter. The decoding accuracy for previous incorrect left and right choices started decaying during the late ITI and was significantly lower than the accuracy for either of the previous correct choices. We further found that the decoding accuracy for previous incorrect right choices was slightly lower than for previous incorrect left choices.

We then asked whether the neural representation of certain trial history contexts was more similar than others and whether the similarity persisted across trial time. The more similar the neural representation of a pair of trial history contexts is, the higher the probability that the decoders misclassify them. We therefore examined the errors made by the decoders by constructing confusion matrices over trial time. We found that the types of misclassifications changed with time. At the very beginning of the ITI, right after the mice had finished reporting their decision, the decoders frequently confounded correct with incorrect choices during both left and right decisions (Fig. 2h, *top*). 500 ms later into the early ITI, all the different trial history contexts were highly decodable (Fig. 2h, *middle*) and finally at the beginning of the stimulus presentation during the new trial decoders were less accurate at distinguishing between previous left and right choice for incorrect trials, whereas previous correct left and right trials were still well decoded (Fig. 2h, *bottom*). These findings suggest that neural representations evolve from initially tracking previous choices to later keeping a record of combinations of previous choices and outcomes during the ITI and the stimulus and action periods.

ACC neurons exhibit long intrinsic- and task-related firing timescales[37–39], making them suitable to retain information over

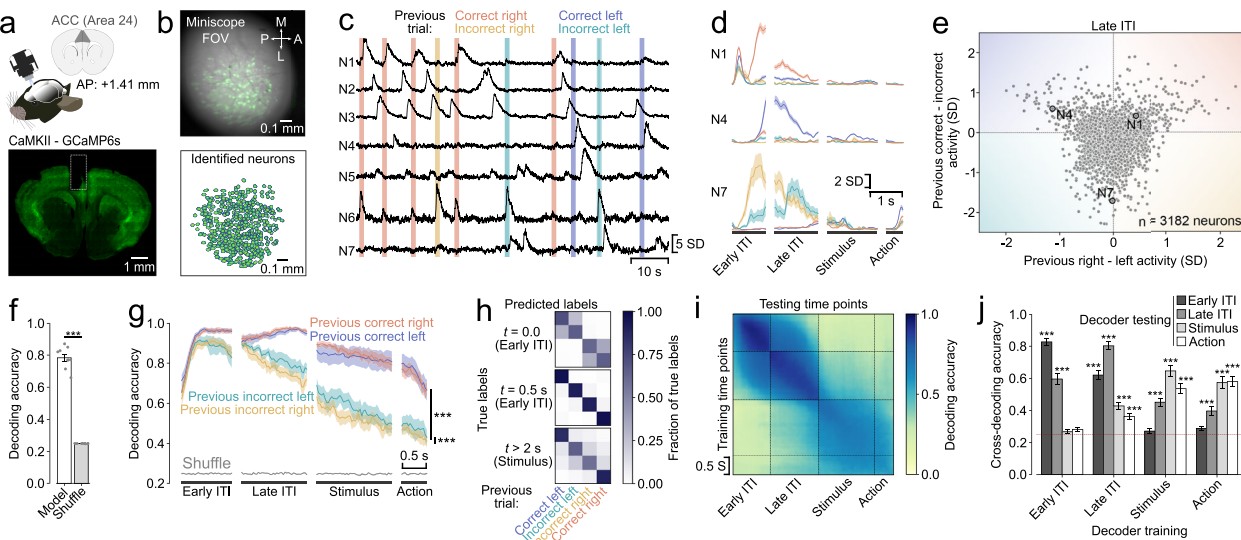

**Fig. 2 | ACC neuron populations uniquely represent multiple trial history contexts.** **a** Transgenic mice expressing GCaMP6s under the control of the CaMKII promoter were implanted with a gradient refractive index lens above their ACC; neural activity was recorded using a miniaturized fluorescence microscope (top). Histological verification of GCaMP6 expression and lens position (bottom). Coronal brain section was adapted from[75]. **b** (Top) Field of view from an example recording. A single frame is overlaid with the variance projection of the entire recording (green). White arrows depict anterior (A)−posterior (P) and medial (M)−lateral (L) axes. We identified individual neurons using an automatic segmentation algorithm[76] (bottom). **c** Example traces from seven simultaneously recorded neurons. Vertical lines represent the first second of the ITI after respective choices and outcomes. **d** Peri-event time histograms of three neurons from (**c**) showing distinct responses for different trial history contexts. Lines and shading represent mean ± sem. **e** Mean activity difference for previous right vs. left choices (*x*-axis) plotted against mean activity difference for previous correct vs. incorrect trials (*y*-axis) during the late ITI for individual neurons from one session per mouse. Circled points mark the example neurons shown in (**d**), color shading schematizes

increasing selectivity for specific trial history contexts. **f** Mean ± sem accuracy of a linear decoder trained to predict trial history based on ACC population activity. Bars represent the average across the entire trial time (*n* = 9 subjects). ***$P < 0.001$, linear mixed-effects model with session nested in subject as random effects, two-sided coefficient testing. **g** Mean ± sem decoder accuracy over trial time for different trial history contexts. ***$P < 0.001$, linear mixed-effects model with session nested in subject as random effects. Coefficient testing was two-sided and *P*-values were adjusted for multiple comparisons. **h** Decoder confusion matrices for three different time points with reference to the last choice. $t > 2$ s is the start of the stimulus presentation. **i** Decoding stability: trial history decoders were trained for every time point and tested on every other time point. The average over all subjects is shown. **j** Mean ± sem cross-decoding accuracy across subjects (*n* = 9) divided by trial phase. Dashed red line reflects chance level. ***$P < 0.001$, linear mixed-effects model with session nested in subject as random effects and post-hoc comparison between model and shuffled accuracies. Coefficient testing was two-sided and *P*-values were adjusted for multiple comparisons.

extended periods of time. We therefore asked whether the neural representations encoding trial history information remained stable[40–42] or whether they evolve gradually[43] across different trial phases. To test this, we trained decoders on all time points and tested their decoding accuracy on all the other timepoints. We found that decoders were very stable within the early and late phases of the ITI and within the stimulus and action phase, with a marked split between ITI and new trial timepoints (Fig. 2i). When we quantified the decoder stability, we found that trial history decoders trained on the early ITI also performed very well on the late ITI time points but were at chance level for the later trial phases (Fig. 2j). Decoders trained on the late ITI predicted trial history better than chance across the entire trial. Finally, decoders trained on the stimulus and action phase performed above chance on all other phases but the early ITI. Neural representations of trial history could weaken over time, or they might get overwritten as animals make their subsequent choices. To test for the influence of time on trial history decoding we separated the trials based on how much time elapsed between the beginning of the early ITI and the initiation of the current trial (ITI duration) and looked at the decoder accuracy in the stimulus phase. We found that the decoding accuracy was significantly higher than chance for ITI durations up to 10 s (Supplementary Fig. 2c) and that the markedly higher decoding accuracy for previous correct trials as compared to previous incorrect trials was mainly due to trials with short ITI durations (Supplementary Fig. 2d). We further found that trial history was still decodable after mice had already made their next choice (Supplementary Fig. 2e, f). Taken together, these findings point to a temporally stable trial history

representation during the ITI that undergoes a major reorganization when the mice transition from the ITI to initiating a new trial.

Finally, we assessed whether trial history could be decoded better in sessions where subjects more strongly relied on trial history to guide their choices. We found no correlation between behavioral trial history strength and trial history decoding accuracy (Supplementary Fig. 4a). There was a weak positive relationship between behavioral weights for trial history and decoding accuracy only for previous incorrect right trials (Supplementary Fig. 4b). Our findings thus suggest that trial history information is similarly encoded in the ACC population activity regardless of the animals' behavioral trial history biases.

### Neural representations of trial history cannot solely be explained by body posture or movements

Recent work in mice has demonstrated that much of the neural activity not only across motor but also sensory cortical areas can be explained by spontaneous movements[24,44–46]. Some movements are highly stereotyped and aligned to task events, task variables or animals' past actions[47]. We therefore examined whether neural signatures of trial history were simply a reflection of the animals' movements or whether they represented a uniquely cognitive signal. To address this question, we fitted linear encoding models which relate the activity of individual neurons to task variables, and postural and movement regressors[24]. The task variable regressors were choice, outcome, trial history and visual stimulus events; movements included the instructed nose pokes into each of three different ports, a lower dimensional representation of either the full behavioral video

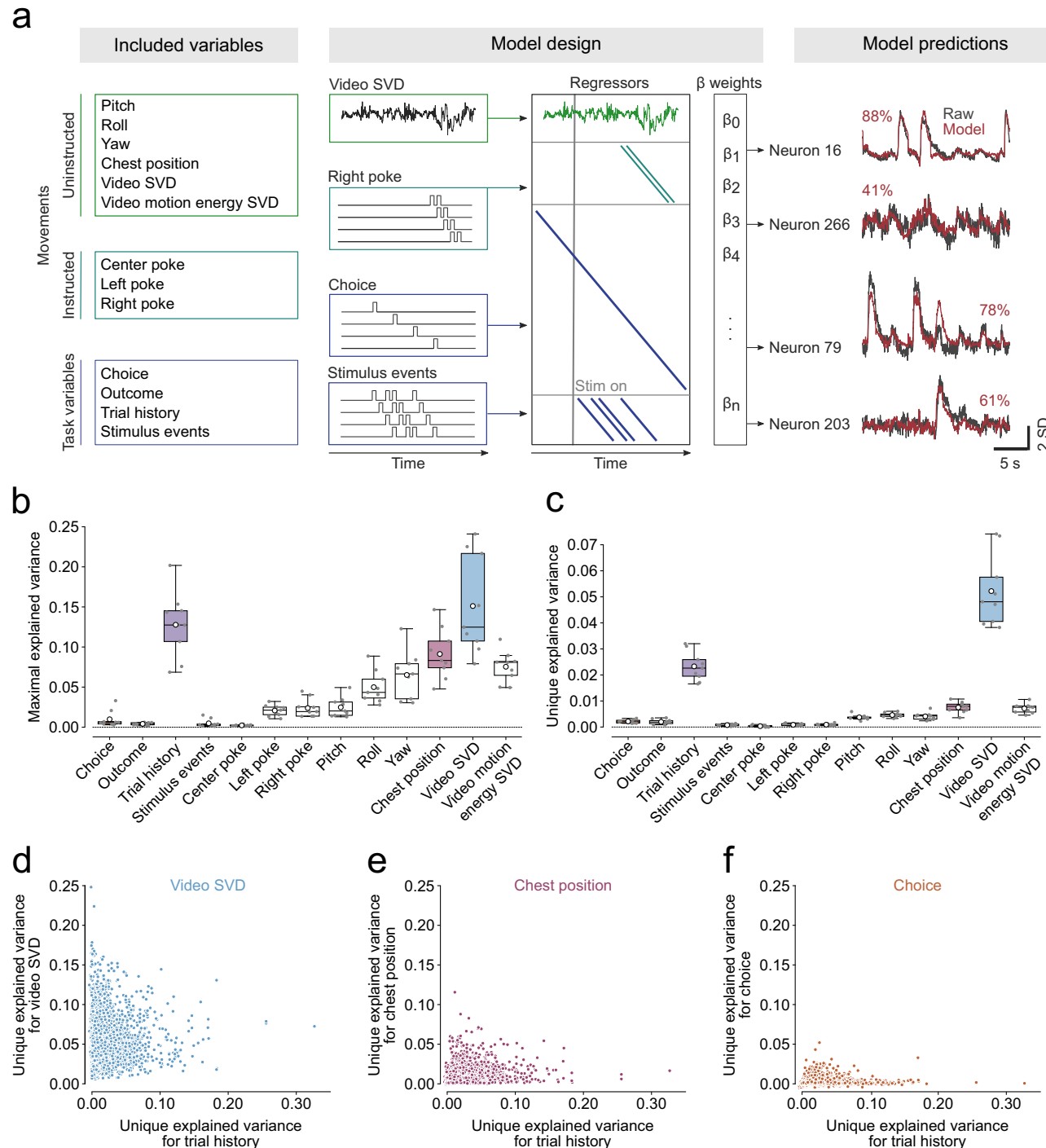

**Fig. 3 | Trial history explains considerable unique variance on the single-cell level. a** Linear encoding model design. Neural activity was modeled as a function of task variables, instructed- and uninstructed movements (left and center). Right panel shows measured (grey) and reconstructed (red) activity of four example neurons. The number indicates the percent explained variance. **b**, **c** Maximal amount of neural variance (**b**) or variance uniquely (**c**) explained by

different regressors. Boxes represent median of subject means ± interquartile range, whiskers show 1.5× the interquartile range, white circle depicts mean and grey scatters individual subjects ($n = 9$ subjects). **d**–**f** Unique variance of all neurons for trial history plotted against the unique variance for Video SVD (**d**), chest position (**e**), and choice (**f**).

(Video SVD) or the motion energy of the video (Motion energy SVD), and neurons' tuning to head-orientation angle and body position (Fig. 3a, *left and center*, Supplementary Fig. 3a). To avoid overfitting the models, we regularized the regressor weights using ridge regression and ran tenfold cross-validation. On held-out trials, the models on average predicted $32.8 \pm 1.9\%$ of the neural variance across the imaged ACC populations with above 85% explained variance for

some individual neurons (Fig. 3a, right, compare red and grey traces for each neuron).

Having built a model that successfully predicted single-trial neural activity, we set out to determine which variables contributed to this prediction. We first asked how much neural variance could be explained by each model regressor alone (maximal explained variance). To address this question, we fitted encoding models with all

regressors shuffled except the regressor of interest plus a time-varying intercept. We then also fitted an intercept only model and subtracted its explained variance from the single-regressor model explained variance. This ensured that the explained variance was driven by the specific variable in question rather than temporal fluctuations shared across all trials. We found that uninstructed movements, particularly the video and video motion-energy regressors, explained relatively high amounts of variance on their own (Fig. 3b). This observation complements previous reports that movements explain considerable neural activity in head-restrained mice[24,44,45] and demonstrates that movements also have a major impact on neural activity in freely moving animals[48,49].

Because task variables can be highly correlated with each other (or with movements) the maximal explained variance is vulnerable to over-estimating the true explained variance (Supplementary Fig. 3b). To account for this, we computed the unique explained variance of the different regressors. To this end, we fitted the full model with only the regressor of interest shuffled and then subtracted its explained variance from the full model variance (Supplementary Fig. 3c). We found that not only Video SVD but also trial history accounted for considerable unique neural variance (Fig. 3c).

This observation is in stark contrast to the impact of other task variables in other areas, where task variables typically account for minimal neural variance and are dwarfed by movement signals[24].

We then assessed the temporal profile of the maximal and unique explained variance trial history. We observed that the maximal explained variance for trial history decreased over the different trial phases, similar to the decrease in trial history decoding accuracy (Supplementary Fig. 3d). In contrast, the unique explained variance for trial history remained relatively stable (Supplementary Fig. 3e).

We further assessed whether the maximal or the unique explained variance for trial history of the ACC neuron populations were related to the behavioral trial history biases. Similar to the decoding results, we did not observe any correlation between behavioral trial history biases and maximal or unique neural variance explained by trial history (Supplementary Fig. 4c, d), providing reassurance that neural trial history encoding signals are not simply driven by behavioral biases. However, we observed that trial history encoding and decodability tended to be lower in females than in males (Supplementary Fig. 5).

Previous work demonstrated that neurons in prefrontal and parietal cortices of primates, rats, and mice have mixed selectivity for different task variables[35,36,50–53]. We thus asked whether multiple regressors might explain variance within the same neurons. Because multiple collinear variables could drive the maximal explained variance, we only considered the more conservative measure of unique explained variance to assess mixed selectivity. We indeed found that in many neurons, where trial history explained unique variance also video SVD and to lesser extent chest position and choice could explain sizeable amounts of unique variance (Fig. 3d–f) pointing to a high degree and diversity of mixed selectivity in ACC neurons.

Taken together, these data demonstrate that ACC neurons strongly encode trial history. In contrast to earlier findings for neurons in sensory and motor areas of the dorsal cortex[24,44,45], the trial history encoding in the ACC cannot simply be explained by subjects' spontaneous or instructed movements.

### Neural dynamics representing trial history are low-dimensional and similar between subjects

Having established a method to estimate neural activity driven by trial history rather than a set of other variables, such as chest position or head-orientation, we then sought to understand whether there is a limited set of neural population dynamics representing trial history shared across many neurons in the ACC. To address this question, we first applied principal component analysis (PCA) to the weights of trial history, the video SVD components or the neuron's tuning to the chest position obtained from the full encoding models. We observed that within the top two dimensions, the four trial history contexts were well-separated (Fig. 4a) while the first and third dimensions of chest point encoding showed well-defined place tuning (Fig. 4b). We then compared the number of dimensions required to capture at least 90% of the variance of the weights for trial history, chest position tuning and video SVD (Fig. 4c). We found that relatively few PCA dimensions were necessary to explain the trial history encoding ($12.0 \pm 0.6$) and chest position ($17.2 \pm 1.7$). By contrast, many more dimensions were needed to explain the same amount of variance in the video SVD ($63.0 \pm 6.4$, Fig. 4d). These findings indicate that trial history information is encoded in a small set of temporal dynamics and that there are far fewer chest tuning motifs in the ACC population than video SVD motifs.

Next, we hypothesized that if the neural dynamics for trial history represented a reliable monitoring process, they should be similar across different sessions from the same subject and even between subjects. To test this hypothesis, we used procrustes analysis[54]. Procrustes analysis compares two geometric shapes with each other by finding a set of matrix transformations that minimize the disparity between the two shapes (Fig. 4e). The more similar two shapes are after minimizing their disparity, the smaller the procrustes distance. When we matched the trial history dynamics across different sessions, we observed very similar shapes (Fig. 4f, *top*), whereas not all dimensions of chest point encoding matched well between two example sessions (Fig. 4f, *bottom*). We then quantified dissimilarity (procrustes distance) of trial history, chest position, and video SVD encoding between sessions from the same subjects and between different subjects (Fig. 4g). We found that the within- and across-subject procrustes distance of trial history encoding was significantly lower than the distance of chest and video SVD tuning. Importantly, there was no difference between within- and across-subject procrustes distance for trial history encoding weights.

These results provide evidence for the hypothesis that trial history dynamics discovered by the linear encoding model might represent a monitoring process shared by different subjects.

## Discussion

We set out to test whether ACC neurons encode behavioral history in fully deterministic perceptual decision-making tasks. To do this, we trained freely moving mice to make decisions about abstract features of visual stimuli and measured neural activity of excitatory ACC neurons using a miniaturized, head-mounted microscope. This approach allowed us to simultaneously record neural dynamics from hundreds of ACC neurons during unrestrained perceptual decision-making. We found that trial history influenced decision-making and that subjects weighed trial history differently in each session (Fig. 1e, f). We observed that the activity of individual ACC neurons was strongly modulated by trial history (Fig. 2c–e). Using linear decoders, we showed that the neural representations of distinct trial history contexts are well separated across the entire trial (Fig. 2g) and that these representations remained stable over several seconds (Fig. 2i, j). We showed that trial history explained large fractions of neural variance on the individual neuron level and that much of this variance could not be accounted for by other correlated task variables or movements (Fig. 3b, c). Finally, we found that trial history encoding is low-dimensional and that neural dynamics representing trial history over trial time are relatively similar across different subjects, unlike representations of posture and movement (Fig. 4f, g).

Neural signals in the ACC are proposed to represent multiple cognitive signals, including surprise, effortful decisions, and contextual-, task rule- or strategy representations[4,6,9,11,14–16,55–58]. We find that ACC neurons in expert mice performing a fully deterministic perceptual task without rule-switches encode trial history more strongly than other task variables. This is remarkable for two reasons.

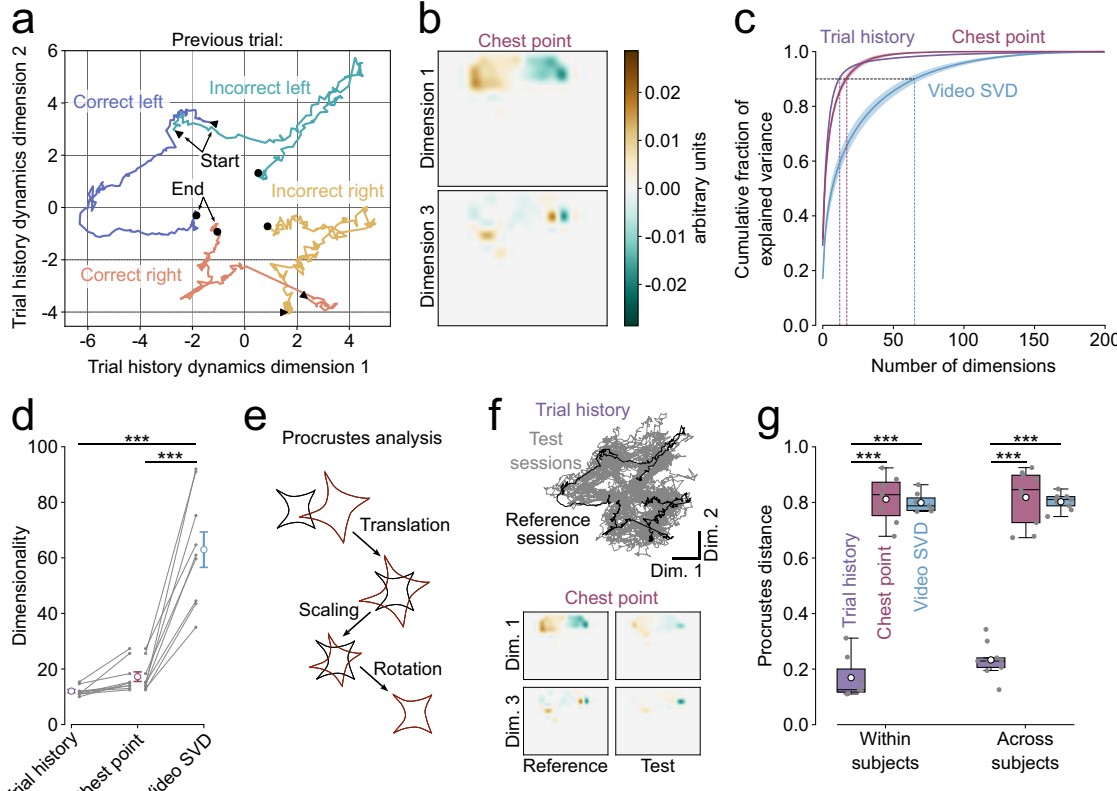

**Fig. 4 | Neural dynamics representing trial history are low-dimensional and similar between subjects. a** Trial history encoding weights projected into the two first principal component dimensions and separated by trial history. Black triangles mark the beginning of the early ITI and circles the end of the action phase. **b** Two dimensions of the chest point encoding weights. **c** Mean ± sem fraction of cumulative explained encoding weight variance for different variables as a function of PCA dimensions. **d** Minimum number of PCA dimensions required to explain 90% of the encoding weight variance. Grey lines represent individual subjects ($n = 9$), with circle and error bar depicts mean ± sem over subjects. ***$P < 0.001$, linear mixed-effects model with session nested in subject as random effects. Coefficient testing was two-sided and $P$-values were adjusted for multiple comparisons. **e** Schematic representation of matrix transformations for procrustes analysis. **f** Trial history dynamics of all other sessions (grey) matched to the reference session (black) on (**a**, top). Two dimensions of chest point encoding weights of an example session (right) matched to the reference session in (**b**, left). **g** Dissimilarity (procrustes distance) of different regressors for different sessions within subjects ($n = 7$) and across subjects ($n = 9$). Boxes show median ± interquartile range across subjects, whiskers indicate 1.5× the interquartile range, white circle depicts mean, and grey scatters individual subjects. ***$P < 0.001$, linear mixed-effects model with random effect of subject. Coefficient testing was two-sided and $P$-values were adjusted for multiple comparisons.

First, the trial structure of our perceptual task was random and lacked a block structure. This ensured that the only informative feature for the animals was the sensory evidence and any reliance on trial history hampered optimal performance. Nevertheless, individual excitatory ACC neurons and the excitatory ACC population faithfully tracked trial history. This suggests that behavioral history monitoring in ACC neurons may be independent of the volatility of the environment rather than emerging as result of having learned the statistics that govern its dynamics. These findings contrast with reports from an earlier study that showed that previous choice and outcome signals only emerge after introducing the first rule reversals in a set-shifting task[18]. However, in their study in head-restrained animals, trials were automatically triggered, and animals licked to report their decisions. In contrast, mice in our task actively initiated trials and reported their choices by freely moving within an arena, adding strong volitional and spatial components to the task and making the behavior costly. It is possible that a combination of these factors may influence trial history representations in the ACC. We further note that Spellman and colleagues recorded from prelimbic rather than ACC neurons, which might explain the divergent observations. Second, a series of recent studies have shown that movement signals explain large fractions of neural variance in many cortical regions of head-restrained[24,44,45] and freely moving rodents[49]. In fact, movements appear to be stronger

predictors of neural activity than internal or cognitive variables across sensory-motor cortex[24,44,45]. Yet, internal states were found to be highly intertwined with the stereotypy of movement patterns[47] and the two might be hard to separate[59–61]. In line with these reports, we find prominent representations of postural- and movement signals in many ACC neurons. However, we also observe strong trial history encoding that was not readily explainable by movements (in contrast to encoding of upcoming choice, for example). Our evidence thus suggests that this encoding might represent an internal cognitive signal.

The trial history encoding in ACC neurons displayed three main characteristics that are well-suited for monitoring behavioral history: rich encoding allowing for flexible readouts of information, temporal stability, and consistency across behavioral biases.

First, many ACC neurons exhibited non-linear mixed selectivity with respect to animals' previous choices and outcomes. Non-linear mixed selectivity supports efficient encoding of task variable combinations in population activity while at the same time enabling flexible read-outs in down-stream regions[35,36] and neurons with non-linear mixed selectivity are wide-spread in frontal[50,52,53] and parietal[51,62,63] cortical regions. In agreement with this idea, we show that all combinations of previous choices and outcomes can be decoded from the ACC population activity without impairing the decoding of previous choice or outcome themselves. The ACC is thus well positioned to

represent different aspects of trial history information and broadcast it to downstream targets.

Our recordings revealed that trial history representations remained stable for at least 1–2 s. We further found that trial history could still be decoded above chance levels after inter-trial intervals that lasted for up to 10 s as well as after the mice had already made another choice. This is reminiscent of reports suggesting that ACC neurons have long intrinsic- and task-related timescales[37–39] that may enable them to retain relevant task variable information into and beyond subsequent trials[16,18] and suggests that they may play an important role in working-memory processes[64,65]. ACC neurons might thus integrate salient information about subjects' actions and their outcomes over multiple trials, as has been shown in secondary motor cortex[66] and accumulate evidence that would suggest a change in the environment[11].

We observed that the trial history representations in ACC were highly consistent and did not change with behavioral biases or across sessions or even subjects. We took advantage of the finding that animals performing our evidence accumulation task showed trial history biases that fluctuated between sessions to probe whether trial history representations would be influenced by these biases. We found that the encoding strength and decodability of trial history from ACC neurons was unaffected by subjects' biases. This indicates that the ACC holds a faithful representation of trial history and might not directly drive biases associated with it. In agreement with this, recent studies in non-human primates and mice reported that ACC and prelimbic activity did not predict choice biases[17] and disruption of ACC activity had no impact on ongoing decision-making[12,13,15,18]. We further found that the dynamics encoding trial history were not only similar across sessions from the same subject but also consistent between subjects, in strong contrast to movement encoding that was highly variable between sessions and subjects. This result lends further support to the hypothesis that ACC neurons track information about previous actions and their outcomes in a generalized form as a cognitive signal that is distinct from highly variable and individualized movement signals.

Accurate monitoring of past choices and outcomes is vital to discover regularities in one's environment early in learning, but it also enables subjects to compare expected with obtained outcomes in dynamic environments to update these expectations if necessary. Our findings propose that the trial history monitoring in ACC provides a substrate for this comparison. The presence of trial history signals in expert mice performing a deterministic perceptual task with a fixed rule may indicate that the animals continue evaluating their task knowledge even after attaining high performance and that they may be poised to explore alternative strategies at any moment[20]. We speculate that the evolutionary advantages of this readiness to switch to different strategies in light of new evidence have greatly outweighed the possible metabolic costs of constantly maintaining an active representation of behavioral history.

Our observations raise a set of questions that can be addressed in future studies. First, we do not know what aspects of the trial history representations might be exclusive to the ACC. Strong trial history signals have also been observed in other prefrontal regions, such as the prelimbic[18] or orbitofrontal cortex[67], the M2[29,66] or the PPC[33,40,63]. It remains unclear whether trial history signals in these regions are similarly independent of movements or of other tracked task variables, and whether they display similar temporal stability and between subject similarity. Second, our approach can only resolve relatively large movements of the full body and limbs of the mice. It is, however, possible that ACC neurons might also encode more detailed orofacial movements and that these movements could account for some of the neural variance for trial history. Future work will have to elucidate to what degree the neural encoding in the ACC is modulated by large full-body versus small facial movements. Third, we have not directly tested the effects of ACC disruption on task performance because

optogenetic intervention would only yield limited new insights. Due to the high non-linear mixed selectivity of ACC neurons for different trial history contexts and their mixing of trial history and movement signals, it is difficult to perturb neural dynamics that are specifically linked to trial history. Indeed, repeated strong optogenetic inhibition of ACC neurons was shown to decrease task engagement rather than influencing subjects' decision-making[68,69]. Fourth, given that head-restrained behavioral tasks have become a staple in circuit neuroscience, it will be crucial to directly compare the trial history and movement signals in the ACC of mice making perceptual decisions in head-fixed and freely moving tasks. This is particularly important because head-restrained tasks may differ from freely moving behavior in aspects that were shown to influence ACC activity, including volitional task engagement, effort, or affective components of discomfort and pain[70]. It thus remains an open question how these task-modality specific factors may influence trial history representations in the ACC.

In summary, our study provides evidence that ACC neurons maintain an internal record of past decisions and outcomes that is separate from physical movements, suggesting that ACC continuously monitors behavioral success even in highly predictable environments.

## Methods
### Animal subjects
We used transgenic mice on a C57/B6 background that expressed the genetically encoded calcium indicator GCaMP6s in calmodulin-kinase II expressing cells under the control of a tetracycline response element (GCaMP6s was constitutively expressed in the absence of doxycycline). We obtained these subjects by crossing TRE-GCaMP6s mice (B6;DBA-Tg(tetO-GCaMP6s)2Niell/J, Jax stock no. 024742)[71] with CaMKII-tTA mice (B6.Cg-Tg(Camk2a-tTA)1Mmay/DboJ, Jax stock no. 007004)[72]. For our study, we used at total of 14 mice of both sexes that had attained expert performance in our visual evidence accumulation task. Two of these animals were excluded because they did not meet our task performance and session metrics (see behavioral analysis below). We therefore included behavioral data from 8 males and 4 females. We obtained high-quality imaging data from 7 of the 8 males and from 2 of the 4 females. For 2 of the 7 males with good imaging quality, we found bilateral damage to the M2 / ACC region in the post-mortem analysis, and we excluded them from the imaging data set. We thus included 5 male and 2 female mice performing visual decisions into the imaging dataset. We further included imaging data from one male and one female performing auditory evidence accumulation that had initially been trained to discriminate visual stimuli. We did not find differences in trial history encoding in these mice and pooled their data with the other mice. For a complete summary, please refer to Supplementary Table 1. Mice were 8–25 weeks old when handling and water restriction started. Mice were housed in a room with controlled humidity (30–70%) and temperature (71–76 °F / 21.67–24.44 °C). Mice were kept on a reverse dark-light cycle, with lights on at 10 am and lights off at 10 pm. Animals were provided with cotton nestlets and received an acrylic shelter and running wheel as additional enrichment. They were fed regular laboratory chow (NIH-31 Modified Open Formula Mouse/Rat Diet 7013) *ad libitum* and given additional treats (Mini Yogurt Drops, Bio-Serv) at the beginning of the day and after performing the task. Animals were water restricted during behavioral training and testing (see below). All behavioral and surgical procedures adhered to the guidelines established by the National Institutes of Health and were approved by the Institutional Animal Care and Use Committee of the University of California, Los Angeles, David Geffen School of Medicine. No statistical methods were used to pre-determine sample sizes. Sample sizes are similar to previous publications.

### Habituation and water restriction
A minimum of one week before the start of the behavioral training, we began habituating the mice to being handled by the experimenters and

let them walk and climb on the experimenters' hands. During the handling animals were also presented with a treat (Mini yoghurt drop). At the same time, we restricted the mice's access to water to 1 ml per subject and day and the water was given right after the mouse handling. To ensure that the subjects maintained healthy hydration levels, we measured their body weight daily during the handling or before running the behavioral task. If their weight fell beneath 80% of their body weight before water restriction, they were supplemented with 0.4 ml of water after the behavioral session[73].

## Perceptual evidence accumulation task

We trained animals on an audio-visual evidence accumulation task[22,23]. In this task mice were required to integrate pulsatile sensory evidence and compare the number of experienced stimuli to an implicit category boundary (12 Hz). Trials were randomly assigned to be high- or low-rate and we did not impose any limitations as to how many times in a row certain trial types could repeat. Mice self-initiated trials by poking into a centrally located port (and thereby breaking an infrared laser beam) and holding their nose inside this port. No timing was imposed on trial initiations and mice were free to initiate as they pleased. After a very brief random delay, this triggered the playback of a train of visual (or auditory stimuli). The train was composed of a set of 15 ms long stimulus events that occurred pseudo-randomly distributed within a duration of 1 s. Stimulus events were required to be separated by at least 25 ms. We generated stimulus trains with 4, 6, 8, 10, 14, 16, 18, and 20 stimulus events, omitting 12 Hz stimulus trains, which would lie on the category boundary. After the stimulus train had finished playing, the same stimulus train was repeated. This ensured that the stimulus did not abruptly terminate as the mice prepared to report their choice. Mice had to remain poking in the central port for at least 1 s (one full stimulus train presentation) plus a random delay drawn from an exponential distribution, after which an auditory go-cue (7 kHz) signaled to them that they could leave the center port and report their choice. The random delay was added to make sure that the animals could not simply time the onset of their choice report and to minimize movement preparation signals in the neural activity before the go-cue. If animals left the central port before the go-cue the trial was judged as an early withdrawal and the mice were presented with a white noise stimulus and 2 s timeout. Animals then reported their decision by poking into one of two ports either located on the left or the right side. If the repetition of the stimulus train was playing at the time of choice report it was immediately terminated. Pokes in the left port after a < 12 Hz stimulus or in the right port after a > 12 Hz stimulus were rewarded with a 4–6 μl drop of water, while left pokes after high-rate trials and right pokes after low-rate trials were punished with a tone (15 kHz) and 2 s timeout. Note that the reward delivery was controlled by the opening and closing of a solenoid valve and most rewards were delivered within under 100 ms. Importantly, the association between stimulus rate and rewarded port remained the same throughout the entire experiment and across all sessions. If animals failed to report a choice within 30 s the trial ended and was deemed a no-choice trial.

All the behavioral testing took place in sound attenuated booths that were illuminated only with infrared light (for video recordings). Animals performed the task inside a custom-made modular transparent red acrylic enclosure (20 × 20 cm, 24 high). Ports were installed along the back wall of the enclosure and separated by two 5 cm dividers made from clear acrylic. The task was implemented using custom written matlab functions and an Arduino-based state machine (Bpod, Sanworks). All auditory and visual cues were delivered via a low-latency soundcard (Fenix, HT Omega) and sounds were played from two speakers located on either side of the central port (Harman-Kardon) while visual flashes were played from two LED panels attached above the center port. Water was delivered to the left and right ports through suspended syringes and silicone tubing and the volume was controlled

via the opening time of a solenoid valve (Lee Company). Animal behavior was filmed through the transparent acrylic floor from underneath the animals (Chameleon3, Teledyne-FLIR, Fujinon 3 MP Varifocal Lens). The videos were acquired at 100 fps and the camera was controlled using labcams (https://github.com/jcouto/labcams). Camera acquisition was synchronized to behavioral events via TTL pulses at the beginning of every trial. All the code to run the behavioral task, and assembly instructions and a parts list are publicly available (https://github.com/churchlandlab/chipmunk).

## Behavioral training procedure

One day before the start of the behavioral training, water restricted mice were habituated to the task enclosure. They were presented with 1 ml water inside the left and right ports and in a weighing boat in front of the ports and allowed to drink and explore the enclosure for 5–10 min. At the beginning of the first full training sessions, the inside of all the three ports was baited with a drop of water. For the first 3–5 sessions, the wait time and all the delays were set to 0 s, so that the mice did not need to fixate inside the central port after triggering the stimulus. Similarly, for these sessions, we removed the dividers between the ports to facilitate exploration, and we set the stimulus to keep repeating until mice made their decision or for up to 30 s (which is the maximum allowed time to choose a response port). We did this to help the mice temporally bind causal task events (center poke -> stimulus, stimulus playing -> response port poke). Once animals exhibited stable center-to-side sequences, we introduced the dividers and started increasing the required center fixation time. For most animals we slowly increased the wait time by 0.3 ms after each successfully completed trial (i.e., no early withdrawal). It took the animals up to 3 months (10.72 ± 1.25 weeks) to reach >1 s wait-times. We also gradually lowered the volume of the water rewards the animals earned after correct trials, starting at 8 μl at the beginning and ending at 4–6 μl at the expert stage. Performance on this task requires learning to wait for a specified period of time, understanding an abstract stimulus-response relationship, and mastering the ability to quickly report choices and appropriately re-initiate trials. We therefore refer to animals that successfully underwent this process as "expert" acknowledging that there is some variability in the final level of performance.

Most of the subjects were exposed to rewards and punishments from the first day of training. This allowed them to learn the task rule (with 4 and 20 Hz stimuli) within 2–3 weeks (10–15 sessions) and avoided the animals developing strong side-biases as has been reported before[23]. One cohort of mice (LO028, LO032, LO051, LY007, and LY008) included in this study was allowed to revise their choices and be rewarded on the correct port after initially selecting the incorrect one during the initial training process. These animals were exposed to punishments after showing the first signs of learning at 24 ± 4 sessions on average. The absence of punishment after incorrect choices, however, was more prone to producing first a switching bias that would have to be counteracted by manipulating the trial structure and present trials rewarded on one side more frequently leading in turn to a side bias. This side bias had then to be counteracted by presenting trials for the other side more frequently before returning to a balanced number of trials on both sides and removing any correlations in the trial structure. This approach, therefore, slowed down the animals' learning of the stimulus rate—side association. We started introducing the more difficult stimulus rates once the animals waited for >1 s and performed at above 80% correct on the easy stimulus rates.

## Behavioral analysis (Fig. 1 and Supplementary Fig. 1)

We included subjects for the behavioral analyses based on a two-step selection process: First, we only included subjects that learned to wait for >1 s (including the random delay) and that experienced all the different stimulus rates at least during one session and that performed above 80% on easiest trials at least once. Second, from these subjects

we included only sessions where the animals actually waited > 1 s plus delay, experienced at least two different pairs of stimulus rates (for example 4–20 and 6–18 Hz), performed at least 100 completed trials and where the early withdrawal rate did not exceed 55%. We deliberately did not filter individual sessions for high performance to capture the session-by-session variability in biases that might affect performance levels. Only animals performing the visual version of the task were included for the behavioral analyses.

We fitted individual psychometric curves for each subject using all the completed trials from all their sessions and then averaged the recovered psychometric functions (Fig. 1c). Psychometric functions were modeled using cumulative Gaussian functions parametrized by four parameters, perceptual bias ($\alpha$), sensitivity ($\beta$), lower ($\gamma$)- and upper lapse rate ($\lambda$):

$$p_{right\ side\ choice} = \gamma + (1 - \gamma - \lambda) \cdot F(x; \alpha, \beta)$$

where $F(x; \alpha, \beta)$ is the cumulative Gaussian function:

$$F(x; \alpha, \beta) = \frac{\beta}{\sqrt{2\pi}} \int_{-\infty}^{x} e^{\frac{-\beta^2 (x-\alpha)^2}{2}}$$

We also fitted separate psychometric functions to trials with different preceding trial history (Fig. 1d). To do this, we only included completed trials that were also directly preceded by a completed trial (as opposed to an early withdrawal). Again, to have sufficient amounts of data to estimate the psychometric parameters, we pooled trials from multiple sessions. We used a custom Python package to fit the four model parameters (https://github.com/jcouto/fit_psychometric.git).

To assess the overall influence of sensory evidence and trial history on animals' decisions and to analyze session-by-session variability, we constructed logistic regression models that sought to predict the mice's choice based on stimulus rate and trial history (Fig. 1e, f). Our models featured a regressor for stimulus rate plus a categorical regressor for each combination of previous choice and outcome (for example, previous correct left choice) and an intercept (sensory bias):

$$ln\left(\frac{p_{right\ side\ choice}}{1 - p_{right\ side\ choice}}\right) = \beta_0 + \beta_1 \cdot x_{Stimulus\ rate} + \vec{\beta}_2 \cdot \vec{x}_{Trial\ history}$$

where $\vec{x}_{Trial\ history}$ is given by the vector:

$$(x_{Previous\ correct\ left}, x_{Previous\ incorrect\ left}, x_{Previous\ incorrect\ right}, x_{Previous\ correct\ right})$$

For the stimulus rate regressor, the original stimulus rates were centered around 0 and scaled to their minimum and maximum. A 4 Hz stimulus was therefore encoded as −1, whereas a 20 Hz stimulus assumed a value of 1 with 0 representing a 12 Hz stimulus. The four trial history regressors were categorical regressors that were set to 1 whenever the trial history matched and were 0 otherwise. In these models, the intercept represents the log odds of choosing the right side during a 12 Hz stimulus in the absence of any trial history. When fitting the models, we balanced the number of left- and right-choice trials by subsampling from the class with more observations. We iteratively sampled observations from the majority class 20 times, making sure to include all the observations with similar frequencies. For each of these 20 rounds of sampling, we applied tenfold cross-validation. Models were fitted with an l2 penalty set to 1. We chose not to optimize this penalty because the penalty influences the model weights. Thus, keeping the penalty the same allowed us to compare the model weights between different subjects and sessions.

To capture the overall influence of trial history on animals' choices, we calculated the length of the vector of the four trial history

weights (history strength, Fig. 1g and h). All these analyses were performed using custom Python code (https://github.com/churchlandlab/chiCa). We also modeled how choices and outcomes from the last and second last trials influenced animals' decisions (Supplementary Fig. 1g). We did this by including a regressor for the choice, the outcome and their interaction for the last (t-1) and second last (t-2) trial instead of the trial history regressors. These main effects were coded as 1 when the choice at trial t-1 or t-2 was to the right side or when the choice was correct and 0 otherwise. The interaction regressor was set to 1 when the choices were to the right and correct and 0 otherwise. The stimulus regressor was modeled the same way as in the trial history model. Note that when modeling the main effects of previous choices and previous outcomes, the intercept term no longer reflects a trial history free bias (at 12 Hz stimulus rate) but rather represents the condition with all dummy variables set to 0, an incorrect left choice at t-1 and t-2 with 12 Hz stimulus presented in this case. We also assessed whether sessions cluster with respect to their behavioral decoding weights from the trial history models (Supplementary Fig. 1f). To this end, we used UMAP (using umap-learn) to find a non-linear embedding to project the session's 6 regressor weights into a 2-dimesional space and plotted all the sessions color-coded by subject (Supplementary Fig. 1f).

## Surgical procedures

For our neural recordings, we targeted an anterior and ventral portion of ACC, which was proposed to be a homolog of the primate Brodmann Area 24[74]. To gain optical access to ACC on the right hemisphere, we implanted 4 × 1 mm gradient refractive index (GRIN) lens centered at the following target coordinates (with respect to Bregma): AP: +1.3 mm, ML: 0.5 mm, VD: −1.3 mm (−1.5 mm from the skull surface).

Before the surgery, animals were subcutaneously injected with meloxicam (2 mg/kg) for analgesia and enrofloxacin (5 mg/kg) as antibiotic treatment. Mice were then first anesthetized in 3% isoflurane before being maintained on 1–1.5% isoflurane throughout the surgical procedures. Body temperature was monitored and maintained close to 37 °C using a feedback-controlled heating pad (PhysioSuite, Kent Scientific). Animals were placed in a stereotaxic frame (Kopf, Stoelting). We cut the hair on the scalp with scissors and subsequently used a hair removal cream (Nair) to remove remaining hair. After disinfecting the scalp, an incision was made and a diamond-shaped part of the skin was removed. We then carefully detached the neck muscles from the skull using forceps and scraped the skull clean with a scalpel. We next applied a layer of tissue adhesive glue (Vetbond, 3 M) around the edges of the skull to seal off any exposed soft tissue. We then marked the position of the lens, drilled three very small craniotomies arranged in a triangle around the lens site, and inserted three micro screws into the skull (Fine Science Tools). We made a circular craniotomy with 1.2 mm diameter to later insert the GRIN lens. The dura underneath the craniotomy was carefully removed with sharp 30-gauge needles. We aspirated tissue using blunt 28- or 30-gauge needles using a custom vacuum suction apparatus. We were careful to aspirate a cylindrical volume rather than a conical one and to create a flat surface at the bottom of the aspirated volume. All excess blood was aspirated before lowering the lens into the craniotomy using a suction-based lens holder. After placing the lens, we cemented it to the skull and anchored it to the three screws using a three-component dental cement (C&B metabond, Parkell). We then built a support structure around the lens using black dental cement (Orthojet, Lang Dental). The tip of the lens was protected using the lid of a 1 ml eppendorf tube glued to the dental cement structure using acrylic glue (Zap-A-Gap). Mice were allowed to recover for 7 days after the surgery, and they were checked daily. On the first 4 days after surgery, they were injected with meloxicam and enrofloxacin.

After a minimum of 14 days animals were implanted with a miniscope baseplate. To this end animals were again anesthetized in 3%

and maintained on 1–1.5% isoflurane anesthesia, and their body temperature was held constant. The Eppendorf lid was detached using a drop of acetone and the lens surface was cleaned with ethanol. To provide additional contrast, we painted the area around the GRIN lens with black nail polish. We then positioned the miniaturized microscope with a baseplate attached above the lens surface. At this point the anesthesia was lowered to 0.5% to decorrelated cortical activity in order to identify the best imaging focal plane. The baseplate was then attached using black dental cement. After the cement had dried, we removed the miniature microscope and covered the baseplate with a protective cap. We monitored the animals' behavior immediately after recovery. This procedure was non-invasive and did not require administration of analgesics or antibiotics.

### Histology

At the end of the experiments, animals were deeply anesthetized with an intraperitoneal injection of pentobarbital (Euthasol) and transcardially perfused with phosphate-buffered saline, followed by 4% paraformaldehyde. Subsequently, the brain was extracted and stored in 4% paraformaldehyde overnight for post-fixation before being stored in phosphate-buffered saline. The brains were then embedded in 4% agarose and cut into 80 μm sections. Sections were mounted and nuclei were stained using DAPI inside the mounting medium (Fluoromount-G with DAPI, Invitrogen). Photomicrographs of GCaMP6s expression and DAPI were taken using a Nikon Eclipse Ti2 microscope equipped with a Yokogawa CSU-22 Spinning Disk unit to enable confocal fluorescence microscopy. Approximate coordinates of the GRIN lens center were obtained by comparing major landmarks on the sections (position of the anterior commissure, presence of corpus callosum) with a standard mouse brain atlas[75] (Paxinos and Franklin's the Mouse Brain in Stereotaxic Coordinates, 4th edition).

### Miniature microscope imaging and image processing

We imaged the neural activity of ACC neurons using a head-mounted miniaturized fluorescence microscope (Cai et al., 2016, UCLA miniscope V4, https://github.com/Aharoni-Lab/Miniscope-v4). For every subject we determined the best focal plane by performing brief recordings at different imaging depths (by adjusting the electrotunable lens). We imaged the same focal plane across all recording sessions. We used excitation powers ranging from 0.22 to 0.37 mW/mm$^2$ and adjusted them to optimize imaging quality. Along with the images, we also acquired data from gyroscopes on the miniaturized microscope and later reconstructed the animals head-orientation angles from these data. Images and gyroscope data were acquired at 30 frames per second. We synchronized the behavioral events and the imaging by logging the TTL pulses from the behavioral state machine, indicating the start of a new trial and TTLs from the miniscope signaling the acquisition of a frame on the same computer using a Teensy (Teensy 3.2).

To segment images and identify individual neurons, we used a custom data handling and processing pipeline (https://github.com/jcouto/labdata-tools) and built a plugin to run caiman-based image segmentation routines within this framework[76] (https://github.com/flatironinstitute/CaImAn). We first spatially down sampled the videos by a factor of 2 using the imresize function in Matlab (which yielded better results than comparable methods in Python). We then corrected the videos for motion along the x and y axes using the normCorre implementation in caiman and subsequently ran the segmentation using constrained non-negative matrix factorization (CNMF-E). Raw traces were temporally detrended with the built-in caiman methods. We then visually inspected the identified putative neurons and excluded signals that were not likely originating from neurons. We used the following exclusion criteria: Location outside of the lens perimeter, aberrant shape and size, absence of activity above local background fluorescence. We then aligned the neural recording to the behavior by identifying the recording frames during which new trials started. At this stage, we also identified frame drops (unexpectedly long intervals between frame acquisition TTLs) and accounted for these by interpolating the neural signals.

### Movement and posture analysis

From the gyroscope data acquired using the miniature microscope, we reconstructed the animals' head-orientation at every imaging time point, which was given by the three angles: pitch, roll, and yaw.

To track the position of the animals in space, we used deeplabcut[77] (https://github.com/DeepLabCut/DeepLabCut). We tracked the following 12 body parts that were visible at the underside of the mice: tail tip, tail center, rectum, genital, chest, nose tip, left foot, right foot, left hand, right hand, left ear, right ear. Although we ultimately only used the chest position for further analysis, we found that increasing the number of tracked body parts increased the tracking accuracy overall. We chose to focus on the chest point as it represents the most central tracked body part (to estimate position in space) and because it was rarely occluded. Models were trained with a total of 160 labeled frames from two different sessions, 80 frames each.

We also generated low-dimensional versions of the full behavioral video. To do this, we used an approximate singular value decomposition (https://github.com/jcouto/wfield) on the mean-centered videos and retained the top 200 components. For every video, we also generated a motion-energy video by subtracting each frame from its preceding one and then taking the absolute value of this difference. We then similarly decomposed these motion energy videos and retained the top 200 components for further analysis.

### Neural decoding analyses (Fig. 2 and Supplementary Fig. 2)

For this study, we were interested in following neural signals that tracked combinations of choices and outcomes from the moment immediately after choice report until late into the next trial, when animals were already reporting their next choice. For the decoding and encoding analyses, we therefore aligned the neural activity to four different trial phases starting with the choice report during the previous trial and ending with the onset of the reporting action during the current trial. These phases were defined as follows: Early ITI: the 1 s period following the choice report; Late ITI: the period lasting from 1 s before to 0.3 s after either the mouse left the rewarded port (after a correct choice) or after the punishment noise and timeout ended (after an incorrect choice); Stimulus: 0.5 s before to 1 s after the onset of the train of visual stimuli; Action: 0.2 s before to 0.3 s after the mice left the center port and started moving towards one of the two response ports.

We decoded the specific trial history context from the neural population activity at the different aligned time points. We fitted individual logistic regression models at each time point using the inferred spike rate signal convolved with a Gaussian kernel (1 frame standard deviation) across trials to decode combinations of previous choice and outcome. (Fig. 2f, g). Hence, each decoder sought to classify the trial history context for different trials using the neural activity at a given time point. The decoders distinguished between the 4 different classes using a one-versus-all approach. We balanced the number of classes in a similar way as described in behavioral analysis, subsampling from the majority classes. We then partitioned the subsampled data into a training set containing 87.5% of the data and a test set containing the remaining 12.5% while maintaining balanced labels in both sets. The activity of individual neurons in both sets was standardized by subtracting the mean of the training set and dividing it by the standard deviation of the training set. This avoided data leakage from the training to the test set. We then first optimized the magnitude of the L2 penalty by finding the value that yielded best model predictions in the test fold within an exponential series from $10^{-10}$ to $10^{12}$. The best parameter was then used to fit the logistic regression models. These procedures were repeated for the other 7 training-test splits.

Finally, this procedure was repeated 20 times using different sub-sampled trials. The final model accuracies represent the averages over all sub-samplings and cross-validation folds. As a control, in parallel, we fitted models where we shuffled the labels yielding chance level decoding accuracy. Decoding analyses to determine the neural decoding of previous choice or previous outcome only rather than all of their combinations, were performed in the same way (Supplementary Fig. 2a). When decoding previous choice or outcome alone, we made sure to equally balance the previous outcomes or choices, respectively, in the datasets. To estimate the trial history decoder performance on the binary problems of decoding only previous choice or outcome, we calculated the fraction of trials correctly classified with respect to the variable of interest, ignoring what the class label was for the irrelevant variable. We further tested whether the accuracy of our trial history decoders during the stimulus phase was influenced by the duration of the ITI, that is, the time that had elapsed between the last choice report and the initiation of the new trial (Supplementary Fig. 2c, d). To achieve this, we binned the trials by their ITI duration and separately calculated the decoding accuracy of the trial history decoders for the trials in each ITI bin. We also fitted decoding models that sought to decode the trial history context of two trials back from the neural activity (Supplementary Fig. 2e, f). These models only included trials for which the two preceding trials were both completed and thus had a defined trial history. These models were fitted the same way the one-trial-back trial history decoders were fitted.

To analyze the similarity of the neural representations between the different classes, we constructed confusion matrices (Fig. 2h). To do this, we counted how many times decoders correctly and incorrectly classified trials as each of the different trial history contexts for every given true trial history. Values are reported as the number of predicted trials divided by the number of true trials for each class.

We also looked at how stable the neural encoding of trial history was on the population level (Fig. 2i, j). We did this by predicting the trial history contexts from neural activity at a given time point using the relationship (the logistic regression model) found at another time point. We did this for every pair of included time points. We made sure to do the predictions with the originally trained models from all folds and sub-samplings and then averaged the prediction accuracies rather than averaging the model coefficients first and then performing the predictions.

### Linear encoding models (Fig. 3 and Supplementary Fig. 3)

To assess what task variables or movements drive neural variance, we constructed a linear encoding model[24]. Importantly, our model was built such that it could not only capture relationships that were fixed across trial time, such as neural tuning for specific locations in the task arena, but also relationships that could vary over time, such as how choice influences neural activity. This was achieved by including kernel regressors that could span parts or the entire aligned trial time (Fig. 3a). These regressors were binary vectors that were set to 1 only at the time of an event. To for example, capture how choice affected neural activity, we added time-shifted versions of these vectors to the model until the time of the events spanned the entire trial time. Variables for which the kernel regressors spanned the entire trial time included choice, outcome, and trial history (one for each combination of previous choice and outcome). Importantly, we also included a trial time variable, which consisted of regressors that were 1 for the same time point across trials. This trial time regressor effectively served as an intercept for the other time-shifted regressors. We also time-shifted regressors that tracked the stimulus events as kernels, so that we could detect neural responses to the stimulus events for up to 0.5 s after the stimuli had occurred. We also included pokes into all the different ports as regressors for instructed movements and allowed them to capture neural variance from 0.5 s before to 1 s after the poke happened. Head-orientation angle tuning was assessed by dividing the full range of 360° into 60 bins (6° per bin). We then built a binary vector whose value was one whenever the animal's head-orientation angle was within that bin and zero otherwise. We did this separately for pitch, roll, and yaw. In an analogous manner, we constructed regressors that tracked whether a mouse's chest was within a spatial bin or not. The bins spanned the entire camera field of view and covered a $1.28 \times 1.28$ cm area. Finally, we included the top 200 video and video motion energy components as analog regressors in the design matrix. To make sure that the video and video motion energy components only contained information about the movements rather than the visual flashes that were also visible on the video, we orthogonalized these regressors against the stimulus regressors using QR-decomposition before fitting the linear models. We then aligned the raw fluorescence traces for all the neurons to the specified trial events in the four different trial phases (as described in decoding analysis).

To fit the models, we used ridge regression, which regularizes the model weights using an L2 penalty. We did this to handle the high degree of multi-collinearity of the different regressors (trial history and chest point, for example) and to counteract over-fitting due to the large number of included regressors. To fit the models, we first split the data into ten folds (see decoding analysis above for detailed description of the procedure), making sure that we equally sampled all trial time points. We then standardized our analog regressors (the video SVD and video motion energy SVD) in the training set and used the mean and standard deviation from the training set to standardize the regressors in the test set. We then standardized the neural activity in a similar fashion. We first determined the optimal regularization strength for each neuron by fitting the models using an exponential scale of regularization strengths from $10^{-3}$ to $10^{5}$ and selecting the value that yielded the highest explained variance in the test set as assessed using the coefficient of determination ($R^2$). We then refitted the models with that value before repeating this procedure in the next fold. Besides fitting the full models, we also fitted single-variable only models, where the regressors for all but one variable and the time-varying intercept were shuffled, and one-removed models, where the regressors of only a single variable were shuffled. We then computed the maximal explained variance per neuron for each variable (Fig. 3b) as the single-variable only variance of that variable minus the trial time-only model variance (representing the intercept). Analogously, we computed the unique variance (Fig. 3c) as the difference between the full model and the one-removed model variances. To reconstruct the explained variance for specific variables over trial time (Fig. 3d), we retained the predicted activity for each neuron of all the folds and then re-ordered them to reflect original trial time. We calculated the coefficient of determination for each time point separately. Note however, that the models were fitted and regularized to explain variance across the entire trial time and so the estimates of explained variance for individual trial times tends to be lower than the full model explained variance.

### Variable dimensionality and procrustes analysis (Fig. 4)

We assessed the dimensionality of the neural representation of specific movements or task variables by separately applying PCA to the encoding model weights (from the full model) of the different variables. We then defined the dimensionality as the minimum number of dimensions required to explain 90% of the weight variance across the neuron population (Fig. 4c, d). We measured the similarity of the population representation of task variables within and between subjects using procrustes analysis (Horrocks, Rodrigues, and Saleem, 2024, Fig. 4e–g). Procrustes analysis seeks to minimize the (least-squares) euclidean distance between two shapes by identifying the optimal matrix translation, scaling and rotation of the target shape:

$$D^2(X_1, X_2) = \|X_2 - \beta X_1 \Gamma - \kappa^T\|^2$$

Where $\|X\| = \{trace(X^T X)\}^{\frac{1}{2}}$ is the eudlidean norm, $X_1$ denotes the shape being matched, $X_2$ the reference shape, $\beta$ is the scale parameter, $\Gamma$ the rotation matrix and $\kappa$ the location vector. To match the shapes of the matrices for procrustes analysis and to ensure that we were considering meaningful low-dimensional dynamics or behavioral tuning motifs, we determined the sessions with maximal dimensionality (i.e., dimensions necessary to explain 90% of the variance) and used this number of dimensions for all the other sessions. We did this for every variable separately. The dissimilarity between shapes was expressed as the procrustes distance, where identical shapes have a distance of 0, whereas a value of 1 indicates maximal dissimilarity.

### Data reporting and statistical analyses

Unless indicated otherwise, data are presented as mean ± standard error of the mean (sem) over subjects. All boxplots show the median ± interquartile range of the data and the whiskers depict 1.5 times the interquartile range. Means are shown on boxplots as white circles. Where multiple sessions were acquired per subject, the sessions were averaged first before plotting the averages over subjects. However, the statistics were done on the original data. Because we repeatedly sampled neural activity from the same field of view, several neurons are likely present in multiple recordings. To avoid representing multiple samples from the same neurons, on figures showing data from individual neurons, we only include the session with the most recorded neurons per mouse.

All statistical comparisons were performed using linear-mixed effects models (lmerTest, R) with subject identity as a random effect. All tests were two-tailed. When multiple measures within a session were analyzed, we included the session identity as a random effect nested within subject. We only included random intercepts in our mixed-effects models. Post-hoc group means were compared by estimating the marginal means from the mixed-model fits using emmeans and we used Sidak's method to correct for multiple comparisons (R). Pearson correlation coefficients were computed using scipy.stats.pearsonr and their significance was assessed using the built-in two-tailed permutation test. Detailed descriptions and results of all the statistical analyses are provided in the Supplementary Data 1.

## Data availability

All the data used in this study have been deposited on figshare[78] and are publicly available under the following accession code: https://doi.org/10.6084/m9.figshare.30670382. Raw data files (calcium imaging movies, behavioral videos) can be made available upon request. Source data are provided with this paper.

## Code availability

All code is publicly available. The code to run the analyses and statistical models, and to generate the figure panels presented in this study, can be found in the following repository[79]: https://github.com/LukasOesch/Oesch_et_al_Trial_History_ACC.git. The code provided in this repository depends on the following other toolboxes: https://github.com/churchlandlab/chiCa and https://github.com/jcouto/fit_psychometric.git. Wherever specific packages were used to pre-process data, the packages are cited in the corresponding "Methods" section.

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

## Acknowledgements

We thank Letizia Ye and Marvin Vasquez for animal training and care, Marvin Vasquez for acquiring the histology photomicrographs, and Anup Khanal for technical help. We thank Pinping Zhao for guidance on tissue aspiration and GRIN lens implantation procedures and Federico Sangiuliano Jimka for technical assistance with the V4 miniscopes. We thank Alicia Izquierdo and Pieter Goltstein for helpful discussion on earlier versions of the manuscript. We would also like to thank all the lab members for their feedback and discussions. This work was supported by NSF-NCS collaborative award (2219946), the NIH R01 EY022979 to A.K.C., and the Swiss National Science Foundation Early Postdoc Mobility fellowship (P2BEP3_200212) to L.T.O.

## Author contributions

L.T.O. and A.K.C. designed the project, L.T.O. and J.C. adapted the behavioral task from a previous version, L.T.O., M.C.T., and D.S. trained animals, L.T.O. performed surgeries and collected imaging data, L.T.O. and J.C. wrote analysis tools, L.T.O. analyzed and visualized the data, L.T.O. and A.K.C. wrote the manuscript with inputs from all other authors.

## Competing interests

The authors declare no competing interests.
