## [Transparent Peer Review file · Nature Communications]

Anterior cingulate neurons combine outcome monitoring of past decisions with ongoing movement signals

Corresponding Author: Dr Anne Churchland

Version 0:

Reviewer comments:

Reviewer #1

(Remarks to the Author)

The authors trained freely moving mice to perform a two choice visual perception decision making task and recorded calcium signals during task performance. Activity in ACC found mixed selectivity for task variables, with trial history prominently featured. They found that ensembles could be used to decode trial history, most accurately during the early ITI, but throughout the trial was above chance. Importantly, trial history signals existed separate from movement related influences, showing that the movement does not control all apparent cognitive signals in ACC. Trial history was low dimensional and highly consistent between sessions and subjects, suggesting common coding mechanisms.

Overall, this is an outstanding paper. It is very well polished. The behavior is well developed and appropriately analyzed. The calcium signals are analyzed in a series of well thought out analyses that all complement each other. These are the cleanest findings of trial history signals in ACC I've come across. The experimental control to isolate from the movement related influences is so very important and this data is very much needed in the field. I find myself with no serious concerns or critiques of this paper. I have a couple of small things that needed edifying, but honestly, this is one of the tightest manuscripts I've ever reviewed. This is just really well put together work that I strongly support for publication.

Minor concerns:

More details are needed for the decoding analysis. How was the model(s) trained? Was it on averaged activity over trials? Was it averaged over the entire time periods and then tested for more discreet intervals? Were models created on all the discreet intervals? I'm trying to understand better was the input data was.

Figure 2J is not discussed in the main text and should be.

Typo L295

(Remarks on code availability)

Reviewer #2

(Remarks to the Author)

Oesch et al tackle the critical question about cortical representations of task variables independent of motor output by recording from the mouse anterior cingulate cortex.

The authors use a perceptual decision-making task where mice are rewarded for classifying tones as "low-" or "high-" frequency by making a choice to the left or the right in a three nose-poke two-alternative forced choice operant task.

The authors make the following important contributions:

Show that task history is encoded in the ACC, building upon a large body of prior work

Show that this task history is not explainable by certain measured motor behaviors in mice—something which I believe is an important finding

Show a low-dimensional task representation

Show a non-linear mixed selectivity

Overall, the study presents a straightforward and important set of observations about trial history without any major need for revision.

(Remarks on code availability)

Reviewer #3

(Remarks to the Author)

In their manuscript, Oesch et al. present the findings of an experiment designed to show that neurons in the rodents anterior cingulate cortex encode trial history in a significant way. In my opinion, there are several aspects in the data presentation that are insufficient to draw judgments regarding the reliability and validity of the results. Additionally uncertain are about the selection for the epochs of the task in which is appropriate to emphasize trial history effects.

Major comments:

1. Given how the results are presented, a significant cause of misunderstanding is the definition of the trial and, somehow, what "previous trial" entails. It is evident from the description that animals react to a Go signal that is given at the conclusion of the stimulus (visual or auditory, of varied frequencies) re-presentation period following a variable-length delay. The animals stay still at the door they selected (left or right) and receive the reward—or do not, if they misidentify the stimulus. Of relevance, it is not stated how long this last epoch of reward delivery lasted.

Nonetheless, it is evident from the definition of "early intertrial" (Early ITI; page 21) that the trial's conclusion is defined when the animal enters the port. The Early ITI lasts one second, regardless of the animal's current behavior, which will differ in successful trials (the animal remains to drink) compared to erroneous trials (the animal probably leaves the door in the absence of a reward). The Late ITI has a noisier definition. For correct trials, it refers to the instant the animal exits the door; for error trials, it refers to the conclusion of the 2-second wait (penalty). One second prior to and three hundred milliseconds following the two distinct events is defined as the Late ITI epoch. It is unclear what occurs in the error situation, although we can assume the animal is probably still licking the reward source in succeeding trials.

Given that the animal can only assess its decision during the early and late ITI epochs, even after reading the manuscript multiple times, I still find it very difficult to comprehend the reasoning behind the decision to analyze the effect of the trial history in the defined ITI epochs. Since the animals are still evaluating what they have just chosen to do, it is obvious that the ITI epochs are not ideal for assessing potential monitoring of the outcome in the prior trial. It would have made more sense to look for memory of the recent past in a 'real ITI' (after task completion) or in the moments before the stimulus occurred, with the animal still at the central door, a state that is behaviorally the same (neutral) regardless of the choice made later.

I am unable to determine the conclusiveness of the analysis conducted considering these important confounding factors. If the authors want to remain consistent to their decisions, one required recommendation is to limit the analyses (for behavior and neural encoding) to current correct trials only.

2. Additional information is required. Table 1 shows the number of imaging sessions, while the same is not true for behavior sessions. This information is crucial, for example, to comprehending the reliability of the data presented in Figure 1, and the validity of the following neural analyses.

The number of sessions in which each animal employed the entire set of eight distinct stimuli (as mentioned on page 17, line 553), must be also reported in the same table. Lastly, kindly include the number of "no-choice trials" and "early withdrawal" trials for each session in this same table or a different one.

3. Data from one specific animal's behavior [blue dots] appear to have a strong impact on the results of Figure 1H (right side). By eliminating the data from this animal, the authors should confirm the accuracy of their claim. It's also critical to comprehend the extent to which the number of stimuli employed during a given session affects the x-axis value (changing the stimulus rate weight).

4. The decision to selectively include data from nine of the eligible animals in Figure 3 requires an explanation from the authors. Which criteria were used for this further exclusion?

5. I have difficulties comprehending the meaning of the last statement on page 9, lines 271-3. The mention of earlier findings in "many regions of the dorsal cortex" requires references. Additionally, the section that ends too abruptly needs more explanation. It refers to "movements" without mentioning (neither was mentioned in the preceding provided details) whether the analyses conducted also included movements that are not important to the task.

6. It is confusing that the term "expert animals" is mentioned so frequently. Not all the animals utilized are experts, as seen by the behavioral data in Figure 1 and the fact that it was occasionally required to categorize the option by providing only two frequencies for comparison. Unless the authors can show that the coding they have emphasized in ACC neurons is

influenced by varying amounts of experience, I suggesting eliminating the reference/term.

7. Please include details of behavioral data for each session/animal utilizing the sound stimuli in the Supplementary Materials.

Minor comments

1. The title of the manuscript needs to be changed. The procedure outlined is less about making decisions and more about a possible monitoring activity.
2. Considering the Go signal's declared presence, please remove the reference to "self-initiated" from page 3, line 78 (and other places in this document).
3. Could you perhaps clarify how the findings in Fig. 1D relate to those in Sup Figs. 1A-D?
4. The epochs shown in Figure 1B should ideally be modified in accordance with the epochs duration specified in the Methods. The "Action" period, for instance, starts 200 ms prior to the observed Movement beginning.
5. Describe the significance of Figure 4A's black triangles and circles.
6. The information in lines 482-4 on page 15 about what occurs during the delay period needs to be clarified. The point at which the stimulus is no longer presented is unclear. According to the text, the animal might change its mind in the movement phase after exiting the main window. Has this kind of conduct been confirmed and removed from the analysis?
7. Could you please clarify if the baseline pool of 100 trials (line 554 on page 17) is referred to by the 55% of withdrawal trials?
8. Kindly update Table S1 to reflect the statement that lens coordinates, are 'approximate' at line 676 (page 20).
9. Could you please clarify if the focal plane was changed every day? (line 684–page 20).
10. Please fix the typo error 'varinace' in D and E in Supporting Figure 3.
11. When discussing the data they analyzed, authors frequently use the term "posture" without providing a definition.

(Remarks on code availability)

Reviewer #4

(Remarks to the Author)

This manuscript builds on previous work from the same lab, which demonstrated that a large proportion of frontal cortical activity can be attributed to the encoding of ongoing spontaneous movement, using careful analyses combining high-speed videography with large-scale neural recordings. Leveraging a similar approach in freely moving mice performing a fixed-rule task, the authors now show that ACC neurons encode previous choice–outcome combinations (trial-history) with nonlinear mixed selectivity, in a manner that is distinct from movement or postural information. This suggests that trial-history encoding in the ACC likely reflects a genuine 'cognitive' signal. Notably, the presence of these trial-history representations even in a fixed-rule task implies that the ACC is wired to constantly track its own performance and strategy, rather than adapting to environmental volatility. Given a series of recent studies describing trial-history encoding and ACC function, these findings are important and timely. The analyses described in this manuscript also help set a new standard for the field. Overall, the paper is well-written and clearly presented, with only relatively minor issues, as outlined below.

(1) The authors claim that the trial-history representation was significantly lower in females. However, according to Supplementary Figure 3, the number of female samples is only three. While I appreciate the effort to report sex as a biological variable, the limited sample size makes this conclusion less reliable. It would be more appropriate to either omit the sentence in lines 260–261 or rephrase it to describe the observation as a trend.

(2) Many recent studies reporting that cortical activity is better explained by movement have used head-fixed mice and focused primarily on orofacial movements (often with an additional camera to monitor the main body). In contrast, the current study analyzes main body movements during free movement, without detailed tracking of orofacial behavior, making direct comparisons challenging due to differences in the body parts and types of movement. Can you rule out the possibility that orofacial movements, which are underrepresented in the current dataset, contribute more strongly to ACC activity than the main body movements? If there are existing references that explain cortical activity in terms of similar body parts/movements, citing them would strengthen the comparison and the overall argument.

(3) At what point were the behavioral and recording data acquired, specifically in relation to the stability of the environment? Since the environment is dynamically adjusted during training to shape behavior (as described in the Methods), it is important to clarify how long the environment had been stable at the time of data collection. While the inclusion criteria for sessions are described on pages 16–17, this specific information appears to be missing. Ideally, it should be stated clearly in the main text as the authors emphasize a non-volatile environment during data acquisition.

Minor comments:

Figure 2J seemed not to be referred to in the main results.

(Remarks on code availability)

Reviewer #5

(Remarks to the Author)

(Remarks on code availability)

Version 1:

Reviewer comments:

Reviewer #1

(Remarks to the Author)

The authors have fully addressed all of my previous concerns.

(Remarks on code availability)

Reviewer #3

(Remarks to the Author)

The authors have answered many of the prior questions and supplied enough information to better understand the approach used to analyse the neuronal and behavioral data. I have no other substantial remarks, but I would like to urge that we make an effort to remove the ambiguity that remains in the definition of trial. In general, it is apparent that a 'current trial' and a 'prior trial' definition exist, although this is not always the case, and there is still some uncertainty (e.g., lines 148; 170; 195-196) of what a trial is. I would also propose revising the notion of 'self-initiated perceptual decision making'. I understand the authors' explanation, in response to my previous comment, but the phrase in this case implies that the decision is self-initiated, which is incorrect.

To clarify the message, I'd want to point up one final source of ambiguity. The definition of ITI has been complicated in the most recent version of the manuscript, used for Supplement Figure 2d. This refers to the classic ITI that separate one trial from another. This is however considerably different from the terms ITI used in 'early ITI' and 'late ITI' elsewhere.

(Remarks on code availability)

Reviewer #4

(Remarks to the Author)

The authors addressed all my concerns.

(Remarks on code availability)

Reviewer #5

(Remarks to the Author)

(Remarks on code availability)

REVIEWER COMMENTS

Summary of response to reviewers: We thank the 5 reviewers who took the time to read our paper and make comments. We were pleased to see so many positive comments from the five referees who read the manuscript. We appreciated their assessment that ours is an “outstanding paper” and “one of the tightest manuscripts I've ever reviewed” and that it is, “without any major need for revision,” and “helps set a new standard for the field”.

We also appreciated the more critical feedback and took these comments to heart as we revised the manuscript. We gave particular consideration to concerns that our choice of time windows for analysis limited our ability to draw strong conclusions from this data. Fortunately, we believe we understand where this confusion came from, and we have taken major steps to explain and justify the analysis windows we selected. Beyond addressing this issue, we have included two supplementary tables, added new text to the Results and Methods, changed the title, and included many new references. We believe that these changes have strengthened the manuscript so that it more clearly supports our original message, that ACC neurons monitor previous choices in a way that cannot be accounted for by body posture or movements.

Reviewer #1 (Remarks to the Author):

The authors trained freely moving mice to perform a two choice visual perception decision making task and recorded calcium signals during task performance. Activity in ACC found mixed selectivity for task variables, with trial history prominently featured. They found that ensembles could be used to decode trial history, most accurately during the early ITI, but throughout the trial was above chance. Importantly, trial history signals existed separate from movement related influences, showing that the movement does not control all apparent cognitive signals in ACC. Trial history was low dimensional and highly consistent between sessions and subjects, suggesting common coding mechanisms.

Overall, this is an outstanding paper. It is very well polished. The behavior is well developed and appropriately analyzed. The calcium signals are analyzed in a series of well thought out analyses that all complement each other. These are the cleanest findings of trial history signals in ACC I've come across. The experimental control to isolate from the movement related influences is so very important and this data is very much needed in the field. I find myself with no serious concerns or critiques of

this paper. I have a couple of small things that needed edifying, but honestly, this is one of the tightest manuscripts I've ever reviewed. This is just really well put together work that I strongly support for publication.

We thank the reviewer for the very positive assessment of our manuscript and we appreciate the minor comments that we will address below.

Minor concerns:

More details are needed for the decoding analysis. How was the model(s) trained? Was it on averaged activity over trials? Was it averaged over the entire time periods and then tested for more discrete intervals? Were models created on all the discrete intervals? I'm trying to understand better what the input data was.

We thank the reviewer for pointing out the incomplete description of methods used for the neural decoding analyses. Individual decoding models were fitted to single trials at individual time points (imaging frames) without averaging. The decoders therefore predicted the trial history context across trials at a given time point using the neural activity at that time point. We have extended the description in the methods section (Lines 786 - 791). It now reads as follows:

“We fitted individual logistic regression models at each time point using the inferred spike rate signal convolved with a Gaussian kernel (1 frame standard deviation) across trials to decode combinations of previous choice and outcome. (Figure 2f, g). Hence, each decoder classified the trial history context for each trial using the neural activity at a given time point. The decoders distinguished between the 4 different classes using a one-versus-all approach.”

Figure 2J is not discussed in the main text and should be.

We thank the reviewer for catching the missing reference to Figure 2j. The main text discusses these findings but lacked the reference to the appropriate figure panel. We have included it in the main text (Line 200).

Typo L295

We have fixed this typo in the revised version.

Reviewer #2 (Remarks to the Author):

Oesch et al tackle the critical question about cortical representations of task variables independent of motor output by recording from the mouse anterior cingulate cortex.

The authors use a perceptual decision-making task where mice are rewarded for classifying tones as “low-” or “high-” frequency by making a choice to the left or the right in a three nose-poke two-alternative forced choice operant task.

The authors make the following important contributions:

Show that task history is encoded in the ACC, building upon a large body of prior work

Show that this task history is not explainable by certain measured motor behaviors in mice—something which I believe is an important finding

Show a low-dimensional task representation

Show a non-linear mixed selectivity

Overall, the study presents a straightforward and important set of observations about trial history without any major need for revision.

We thank the reviewer for the positive evaluation.

Reviewer #3 (Remarks to the Author):

In their manuscript, Oesch et al. present the findings of an experiment designed to show that neurons in the rodents anterior cingulate cortex encode trial history in a significant way. In my opinion, there are several aspects in the data presentation that are insufficient to draw judgments regarding the reliability and validity of the results. Additionally uncertain are about the selection for the epochs of the task in which is appropriate to emphasize trial history effects.

We appreciate the reviewer’s critical assessment of our work. We believe we have addressed these points (below) and that these revisions strengthen the manuscript.

Major comments:

1. Given how the results are presented, a significant cause of misunderstanding is the definition of the trial and, somehow, what "previous trial" entails. It is evident from the description that animals react to a Go signal that is given at the conclusion

of the stimulus (visual or auditory, of varied frequencies) re-presentation period following a variable-length delay. The animals stay still at the door they selected (left or right) and receive the reward—or do not, if they misidentify the stimulus. Of relevance, it is not stated how long this last epoch of reward delivery lasted. Nonetheless, it is evident from the definition of "early intertrial" (Early ITI; page 21) that the trial's conclusion is defined when the animal enters the port. The Early ITI lasts one second, regardless of the animal's current behavior, which will differ in successful trials (the animal remains to drink) compared to erroneous trials (the animal probably leaves the door in the absence of a reward). The Late ITI has a noisier definition. For correct trials, it refers to the instant the animal exits the door; for error trials, it refers to the conclusion of the 2-second wait (penalty). One second prior to and three hundred milliseconds following the two distinct events is defined as the Late ITI epoch. It is unclear what occurs in the error situation, although we can assume the animal is probably still licking the reward source in succeeding trials.

Given that the animal can only assess its decision during the early and late ITI epochs, even after reading the manuscript multiple times, I still find it very difficult to comprehend the reasoning behind the decision to analyze the effect of the trial history in the defined ITI epochs. Since the animals are still evaluating what they have just chosen to do, it is obvious that the ITI epochs are not ideal for assessing potential monitoring of the outcome in the prior trial. It would have made more sense to look for memory of the recent past in a 'real ITI' (after task completion) or in the moments before the stimulus occurred, with the animal still at the central door, a state that is behaviorally the same (neutral) regardless of the choice made later.

I am unable to determine the conclusiveness of the analysis conducted considering these important confounding factors. If the authors want to remain consistent to their decisions, one required recommendation is to limit the analyses (for behavior and neural encoding) to current correct trials only.

We thank the reviewer for raising this important point. Indeed, a clear definition of the task phases and how they relate to the animals' behavior is crucial to interpret our results. First, the "go" cue: this occurred after the first 1 s stimulus period plus a brief, variable delay. Thus, the animals were free to move during the stimulus re-presentation. We have clarified this in the methods section (Lines 516), which now reads:

"Mice had to remain poking in the central port for at least 1 second (one full stimulus train presentation) plus a random delay drawn from an

exponential distribution, after which an auditory go-cue (7 kHz) signaled to them that they could leave the center port and report their choice."

Once the animals poked into the response port, the stimulus immediately ended and the outcome was triggered. In the case of a correct choice, a solenoid valve opened to deliver a water reward. Note that the valve opening time was very brief and rarely exceeded 100 ms (for reward volumes of 4 -5 ul), constraining the window of reward delivery to a rather brief moment, during and after which animals were free to lick for as long as they wished. In the case of an incorrect choice a punishment noise was played for 2 seconds and no new trials could be initiated by the mice by poking into the center port. We have added a sentence in the methods that details the method of reward delivery (Lines 528 - 530):

"Note that the reward delivery was controlled by the opening and closing of a solenoid valve and most rewards were delivered within under 100 ms."

Now, on to the issue of the early and late ITI. We believe there was a misunderstanding here: our original description of the task events seems to have given the impression that the early and late ITI only include moments when the animals are still evaluating what they have just chosen to do. Indeed, this would greatly limit our conclusions. Fortunately, our task design, and our selection of analysis epochs, includes many time points, including those long after the animals have experienced the immediate consequences of their choice. We hope the below figure (Reviewer Figure 1) will be helpful here.

Reviewer Figure 1. Definition of Early and Late ITI for two correct trials (1 & 2) and two incorrect trials (3 & 4)

The early ITI is defined by the moment that the outcome becomes clear to the animal: either a brief (~100 ms) opening of the solenoid to deliver water, or the onset of the punishment tone (Reviewer Figure 1, arrows at left). The duration of this Early ITI was always 1000 ms, regardless of whether a reward was delivered. The Late ITI was aligned to a different event, which by necessity had to differ for correct and incorrect trials. For correct trials, this epoch was aligned to the moment that the animal left the chosen (and thus rewarded) port (top rows, arrows at right); for incorrect trials this was aligned to the moment that the punishment tone ceased (bottom rows, arrows at right). The reviewer is right that there is a subtlety here: for correct trials, the time to leave the chosen port is at the animal's discretion, and occasionally they did this rather quickly (Example 2). This did lead to an occasional overlap between the two epochs (see 2.), an issue we have dealt with via an additional reviewer figure: to ensure that our decoding and encoding results are not influenced by information "bleedthrough" from other task phases we re-analyzed the data excluding these trials. The results of our analyses after excluding any trials with overlap between the Early and Late ITI were very similar to our original results (Reviewer Figure 2).

For incorrect trials, this wasn't an issue because the punishment tone always ceased after exactly 2 seconds (compare examples 3 and 4). In all cases, the final 300 ms of this epoch includes exclusively moments in which the consequences of the previous

decision have ended, and the animal is free to contemplate (or not) what happened on that trial.

Reviewer Figure 2. Decoding and encoding results without overlap between Early ITI and Late ITI. (a) Mean \pm sem accuracy of a linear decoder trained to predict trial history based on ACC population activity. Bars represent the average across the entire trial time ($n = 7$ subjects). $***P < 0.001$, linear mixed-effects model with session nested in subject as random effects. (b) Mean \pm sem decoder accuracy over trial time for different trial history contexts. $***P < 0.001$, linear mixed-effects model with session nested in subject as random effects. (c) Decoder confusion matrices for three different time points with reference to the last choice. $t > 2$ s is the start of the stimulus presentation. (d) Decoding stability: trial history decoders were trained for every time point and tested on every other time point. The average over all subjects is shown. (e) Mean \pm sem cross-decoding accuracy divided by trial phase. Dashed red line reflects chance level. $***P < 0.001$, linear mixed-effects model with session nested in subject as random effects and post-hoc comparison between model and shuffled accuracies. (f and g) Maximal amount of neural variance (f) or variance uniquely (g) explained by different regressors. Boxes represent median of subject means \pm interquartile range, white circle depicts mean and grey scatters individual subjects ($n = 7$ subjects). (h, i and j) Unique variance of all neurons for trial history plotted against the unique variance for Video SVD (h), chest position (i), and choice (j). Note that after removing all the trials where Early and Late ITI overlap the sessions from two animals were excluded and thus the number of subjects dropped to 7.

We fully agree that it is critical for readers of our manuscript to understand these epochs. We have therefore clarified these definitions in the legend to Figure 1b;

"(b) We divided trials into four distinct phases: Early ITI: the 1 s period following the animal's choice report, Late ITI: 1 s before to 0.3 s after

leaving the chosen port after correct choices or after the end of the punishment sound and timeout after incorrect choices, ..."

and in the methods section (Lines: 774 - 784):

"For this study, we were interested in following neural signals that tracked combinations of choices and outcomes from the moment immediately after choice report until late into the next trial when animals were already reporting their next choice. For the decoding and encoding analyses, we therefore aligned the neural activity to four different trial phases starting with the choice report during the previous trial and ending with the onset of the reporting action during the current trial. These phases were defined as follows: Early ITI : the 1 s period following the choice report; Late ITI: the period lasting from 1 s before to 0.3 s after either the mouse left the rewarded port (after a correct choice) or after the punishment noise and timeout ended (after an incorrect choice); Stimulus: 0.5 s before to 1 s after the onset of the train of visual stimuli; Action: 0.2 s before to 0.3 s after the mice left the center port and started moving towards one of the two response ports."

2. Additional information is required. Table 1 shows the number of imaging sessions, while the same is not true for behavior sessions. This information is crucial, for example, to comprehending the reliability of the data presented in Figure 1, and the validity of the following neural analyses.

The number of sessions in which each animal employed the entire set of eight distinct stimuli (as mentioned on page 17, line 553), must be also reported in the same table. Lastly, kindly include the number of "no-choice trials" and "early withdrawal" trials for each session in this same table or a different one.

We thank the reviewer for pointing out the missing information about the mouse behavior. We have expanded our reporting on animal behavior in the supplementary material. There are now two supplementary tables. Supplementary table 1 reports the performance metrics for all the subjects that were included in the analyses in Figure 1 and Supplementary Figure 1. The table shows the subject identity, sex, stimulus modality, the number of sessions included in Figure 1, the number of sessions that included the full stimulus set [4, 6, 8, 10, 14, 16, 18, 20 Hz], the average performance on the easiest trials, the number of valid trials (trials with a choice report that were also preceded by trials with a choice rather than an early withdrawal), the early withdrawal rate, the number of no-choice trials, and the average history strength.

Supplementary table 2 provides more detailed information about all the included imaging sessions, such as: animal id, sex, stimulus modality, approximate lens coordinates, session identifier, mean performance on easiest, presented stimulus rates, mean

performance on easiest, early withdrawal rate, no-choice rate, history strength and the number of recorded neurons.

3. Data from one specific animal's behavior [blue dots] appear to have a strong impact on the results of Figure 1H (right side). By eliminating the data from this animal, the authors should confirm the accuracy of their claim. It's also critical to comprehend the extent to which the number of stimuli employed during a given session affects the x-axis value (changing the stimulus rate weight).

The data points of the animal shown in blue (LY008) indeed lie on the high end of the distribution of trial history strengths. Running the analysis with the data from this subject excluded slightly decreased the strength of the negative correlation (from -0.641 to -0.606) so it remained highly significant (Reviewer Figure 3a). We further split the dataset into sessions where only relatively easy stimulus frequencies were presented (4, 6, 18, 20 Hz, Reviewer Figure 3b) and sessions that also contained stimulus rates closer to the category boundary (Reviewer Figure 3c). We found a wide range of different history strengths - stimulus weight differences for both conditions and both of the analyses showed a highly significant negative correlation between the history strength - stimulus weight difference and the mice's performance (Reviewer Figure 3b, Pearson's correlation coefficient: -0.733, Reviewer Figure 3c, Pearson's correlation coefficient: -0.593, $***P < 0.001$).

Reviewer Figure 3: The difference between history strength and stimulus weight is negatively correlated with behavioral performance. (a) Scatter plot showing the history strength - stimulus weight difference versus the fraction of correct trials for all subjects except LY008. The line represents the fit of the data with a linear mixed-effects model with subject as a random effect. (b and c) Same relationship as on (a) with LY008 included but exclusively for sessions with only easy stimuli (4, 6, 18, 20 Hz, b) or for sessions also including more difficult stimulus rates (c). Colors represent the same subjects across all panels. $***P < 0.001$, Pearson's correlation coefficient.

4. The decision to selectively include data from nine of the eligible animals in Figure 3 requires an explanation from the authors. Which criteria were used for this further exclusion?

We thank the reviewer for requesting this clarification. From the total of 16 mice, 2 were not implanted with a GRIN lens (LO090 and LO091), in 3 mice the imaging quality was

very poor yielding almost no identifiable neurons (LO061, LO071, LY007), and in 2 mice, M2 and ACC were damaged bilaterally during the lens implant (LO037, LO038) and the mice were thus excluded. All the neural analyses therefore included the 9 remaining subjects. We have added a sentence in the methods that describes the criteria chosen to exclude these subjects (Line 468 - 471):

“For 2 of the 7 males with good imaging quality we found bilateral damage to the M2 / ACC region in the post-mortem analysis, and we excluded them from the imaging data set. We thus included 5 male and 2 female mice performing visual decisions into the imaging dataset.”

5. I have difficulties comprehending the meaning of the last statement on page 9, lines 271-3. The mention of earlier findings in "many regions of the dorsal cortex" requires references. Additionally, the section that ends too abruptly needs more explanation. It refers to "movements" without mentioning (neither was mentioned in the preceding provided details) whether the analyses conducted also included movements that are not important to the task.

We fully agree. To address this, we have clarified that statement in the text and added references, placing our findings in context with previous work showing that neural activity in many dorsal cortex regions is largely driven by movements rather than task variables, such as trial history. Our model featured both instructed and uninstructed movements (see schematic on Figure 3A): instructed (or task-related) movements are the nose pokes into each port; all the other movement regressors are uninstructed since the animals were not trained to make these movements. We have made this more explicit in the main text (Line 287 - 290):

“In contrast to earlier findings for neurons in sensory and motor areas of the dorsal cortex^{24,45,46}, the trial history encoding in the ACC cannot simply be explained by subjects’ spontaneous or instructed movements.”

6. It is confusing that the term "expert animals" is mentioned so frequently. Not all the animals utilized are experts, as seen by the behavioral data in Figure 1 and the fact that it was occasionally required to categorize the option by providing only two frequencies for comparison. Unless the authors can show that the coding they have emphasized in ACC neurons is influenced by varying amounts of experience, I suggest eliminating the reference/term.

We use the term expert animals to refer to mice that have reached the full wait time, have experienced all the different stimulus strengths at least once and have had at least one session where the performance exceeded 0.8 (please see Behavioral

analysis section in the methods). We deliberately did not impose a performance criterion on individual sessions because this allowed us to sample from a set of sessions that showed a sizable amount of variability with respect to performance and history strength (indeed, sessions with too few errors are difficult for decoding analyses in which correct and error trials need to be balanced). Thus, it is possible for mice to have an average performance on the easiest trials that is below 0.8. As can be seen on the new version of the Supplementary Table 1 though, most subjects' average performance is higher than 0.8. As for the stimulus difficulty, in fact, for the sessions shown on Figure 1, none of the animals only experienced the easiest stimuli (4 and 20 Hz); all sessions featured at least two pairs of stimulus rates (see Behavioral analysis).

During the imaging experiments the two auditory mice performed poorly on all sessions and were shown only the easiest stimuli during a few sessions (see Supplementary Table 2). However, the majority of mice (n = 7) included were strong visual performers that discriminated between multiple pairs of stimulus rates. Still, we acknowledge the reviewer's concern with our usage of the term "expert" and certainly we do not wish to overstate the subjects' abilities. We have therefore added text in the methods explaining how we use the expression "expert" throughout the manuscript (Lines 568 - 572):

"Performance on this task requires learning to wait for a specified period of time, understanding an abstract stimulus-response relationship, and mastering the ability to quickly report choices and appropriately re-initiate trials. We therefore refer to animals that successfully underwent this process as "expert" acknowledging that there is some variability in the final level of performance."

7. Please include details of behavioral data for each session/animal utilizing the sound stimuli in the Supplementary Materials.

We thank the reviewer for this helpful suggestion. We have included a set of behavioral metrics for all the imaged mice on Supplementary Table 2.

Minor comments

1. The title of the manuscript needs to be changed. The procedure outlined is less about making decisions and more about a possible monitoring activity.

We agree that activity monitoring is a key component of our results. Therefore, we have changed the title of our manuscript to: "Anterior cingulate neurons combine outcome monitoring of past decisions with ongoing movement signals"

2. Considering the Go signal's declared presence, please remove the reference to "self-initiated" from page 3, line 78 (and other places in this document).

We believe there might be some confusion about the order of events here. The mice first self-initiate a trial at any moment that suits them. There is no go-cue at this time instructing them to initiate. Once they have initiated a trial, they must remain in the center port for ~1000 ms (the stimulus plus a very brief delay). After this time has passed, a go cue is presented. We include a go cue at this time because many mice find the long (~1000 ms) wait challenging. The go cue informs the mice that the wait is over, and they can now "go" to report their decisions. To address this issue, we have added a new sentence in the methods section (Lines 506 - 508):

"Mice self-initiated trials by poking into a centrally located port (and thereby breaking an infrared laser beam) and holding their nose inside this port. No timing was imposed on trial initiations and mice were free to initiate as they pleased."

3. Could you perhaps clarify how the findings in Fig. 1D relate to those in Sup Figs. 1A-D?

We thank the reviewer for bringing up this important point. We have included a sentence in the main text that explains that the psychometric fits yield the four parameters reported on Supplementary Figure 1A-D (Lines 101 - 104):

"To test which aspects of behavior were particularly influenced by trial history, we obtained estimates for the perceptual bias, sensitivity, upper- and lower lapse rates for all the mice by fitting psychometric curves to their session performance. We found that trial history only significantly changed the subjects' perceptual bias (Supplementary Figure 1a - d)."

4. The epochs shown in Figure 1B should ideally be modified in accordance with the epochs duration specified in the Methods. The "Action" period, for instance, starts 200 ms prior to the observed Movement beginning.

We thank the reviewer for this suggestion. We have adapted Figure 1b so that the events depicted with arrows are aligned to the timing when they occur during the

different task phases. We have also adjusted the width of the rectangles representing the task phases to reflect their duration (with respect to the other phases).

5. Describe the significance of Figure 4A's black triangles and circles.

We thank the reviewer for pointing out this oversight. We have added the following sentence to the legend of Figure 4: "Black triangles mark the beginning of the early ITI and circles the end of the action phase."

6. The information in lines 482-4 on page 15 about what occurs during the delay period needs to be clarified. The point at which the stimulus is no longer presented is unclear. According to the text, the animal might change its mind in the movement phase after exiting the main window. Has this kind of conduct been confirmed and removed from the analysis?

We have indeed observed that some mice pause briefly after getting out of the center port or that they change their heading direction after having started their reporting action, reminiscent of changes of mind. We have not further analyzed this behavior since it seems more related to the upcoming choice rather than the trial history. Fortunately, our encoding model is designed to identify neural activity that is linked to the changes in head-orientation or body position and distinguish these from signals for upcoming choice.

More generally, the animals are required to hold their snout inside the center port during the brief delay period; they cannot start their reporting action until the go-cue. After the end of the 1 s fixation, the stimulus train immediately repeats. If the animal reports its decision within the second presentation of the stimulus train, the stimulus train terminates immediately when the mouse reports its decision by poking into one of the two side ports. If the mouse does not report within that time the stimulus train simply ends and no more stimuli are played to the animal. We have added some clarification about center fixation and the stimulus repetition into the methods section (Lines 505 - 506, 513 - 514, 524 - 525):

"Mice self-initiated trials by poking into a centrally located port (and thereby breaking an infrared laser beam) and holding their nose inside this port. No timing was imposed on trial initiations and mice were free to initiate as they pleased."

"After the stimulus train had finished playing, the same stimulus train was repeated."

"If the repetition of the stimulus train was playing at the time of choice report it was immediately terminated."

7. Could you please clarify if the baseline pool of 100 trials (line 554 on page 17) is referred to by the 55% of withdrawal trials?

We thank the reviewer for this clarification. We required the animals to have completed 100 trials; we considered early withdrawal trials as incomplete. The sessions therefore include 100 completed choices. We have clarified this in the manuscript (Line 597):

"Second, from these subjects we included only sessions where the animals actually waited > 1 s plus delay, experienced at least two different pairs of stimulus rates (for example 4 -20 and 6 - 18 Hz), performed at least 100 completed trials and where the early withdrawal rate did not exceed 55%."

8. Kindly update Table S1 to reflect the statement that lens coordinates, are 'approximate' at line 676 (page 20).

We have changed the labeling on the new Supplementary table 2 to indicate that the lens coordinates are approximate.

9. Could you please clarify if the focal plane was changed every day? (line 684–page 20).

We recorded from the same focal plane for every recording session and did not change the focus between sessions. We have changed the sentence in the manuscript to clarify this point (Line 728):

"We imaged the same focal plane across all recording sessions."

10. Please fix the typo error 'varinace' in D and E in Supporting Figure 3.

We have fixed this typo in the revised version.

11. When discussing the data they analyzed, authors frequently use the term "posture" without providing a definition.

We have now defined our usage of term posture in the introduction (Lines 67 - 68):

"... tracking animals' posture (position in space and head-orientation angles) and movements."

Reviewer #4 (Remarks to the Author):

This manuscript builds on previous work from the same lab, which demonstrated that a large proportion of frontal cortical activity can be attributed to the encoding of ongoing spontaneous movement, using careful analyses combining high-speed videography with large-scale neural recordings. Leveraging a similar approach in freely moving mice performing a fixed-rule task, the authors now show that ACC neurons encode previous choice–outcome combinations (trial-history) with nonlinear mixed selectivity, in a manner that is distinct from movement or postural information. This suggests that trial-history encoding in the ACC likely reflects a genuine ‘cognitive’ signal. Notably, the presence of these trial-history representations even in a fixed-rule task implies that the ACC is wired to constantly track its own performance and strategy, rather than adapting to environmental volatility. Given a series of recent studies describing trial-history encoding and ACC function, these findings are important and timely. The analyses described in this manuscript also help set a new standard for the field. Overall, the paper is well-written and clearly presented, with only relatively minor issues, as outlined below.

We thank the reviewer for the positive assessment of our manuscript and for bringing up some very interesting and important points that we will discuss below.

(1) The authors claim that the trial-history representation was significantly lower in females. However, according to Supplementary Figure 3, the number of female samples is only three. While I appreciate the effort to report sex as a biological variable, the limited sample size makes this conclusion less reliable. It would be more appropriate to either omit the sentence in lines 260–261 or rephrase it to describe the observation as a trend.

We agree with the reviewer that the sample size for the females is very low. We have changed the sentence (now on Line 277) to be more tentative:

“However, we observed that trial history encoding and decodability tended to be lower in females than in males (Supplementary Figure 5)”

(2) Many recent studies reporting that cortical activity is better explained by movement have used head-fixed mice and focused primarily on orofacial movements (often with an additional camera to monitor the main body). In contrast, the current study analyzes main body movements during free movement, without

detailed tracking of orofacial behavior, making direct comparisons challenging due to differences in the body parts and types of movement. Can you rule out the possibility that orofacial movements, which are underrepresented in the current dataset, contribute more strongly to ACC activity than the main body movements? If there are existing references that explain cortical activity in terms of similar body parts/movements, citing them would strengthen the comparison and the overall argument.

We thank the reviewer for raising this very interesting point! Indeed, our approach cannot capture orofacial movements and is restricted to relatively large, full-body or limb movements. Due to this lack of resolution for fine movements we cannot exclude the possibility that such movements may strongly contribute to the ACC activity and may explain part of the task variable encoding we observed.

The literature on fine movement encoding in the ACC is scarce. One study used a similar encoding model approach as ours to predict neural activity using task variables and facial/body movements in the ACC of mice performing a head-fixed visual discrimination task (Reinert *et al.*, 2021). The authors found that the maximal and unique explained variance was similar for task variables and movements. We note however, that the amount of explained variance was generally relatively low and that the encoding model in this case only featured a few movement / postural regressors. Other studies that focused on more detailed analyses of facial movements during decision-making and recorded activity from neurons in M2 and ALM have reported somewhat contradictory accounts as to whether task variable encoding can or cannot be explained by movements in the orofacial region (Cazettes *et al.*, 2025; Hasnain *et al.*, 2025). On the other hand, recent evidence suggests that full-body movements strongly influence activity across many regions of the dorsal cortex in freely foraging rats (Mimica *et al.*, 2023), similar to what has originally been reported in head-fixed mice. It thus remains unclear generally, how strongly orofacial movements contribute to neural variance as compared to full-body movements and specifically whether fine facial movements could be particularly strongly encoded in the ACC. We have added some discussion about these points in the main text (Lines 366 - 374 and 432 - 437):

“Second, a series of recent studies have shown that movement signals explain large fractions of neural variance in many cortical regions of head-restrained^{24,45,46} and freely moving rodents⁵⁹. In fact, movements appear to be stronger predictors of neural activity than internal or cognitive variables across sensory-motor cortex^{24,45,46}. Yet, internal states were found to be highly intertwined with the stereotypy of movement patterns⁴⁸ and the two might be hard to separate⁶⁰⁻⁶². In line with these reports, we find prominent representations of postural- and movement signals in many ACC neurons.”

“Second, our approach can only resolve relatively large movements of the full body and limbs of the mice. It is, however, possible that ACC neurons might also encode more detailed orofacial movements and that these movements could account for some of the neural variance for trial history. Future work will have to elucidate to what degree the neural encoding in the ACC is modulated by large full-body- versus small facial movements.”

Cazettes, F. *et al.* (2025) ‘Facial expressions in mice reveal latent cognitive variables and their neural correlates’, *Nature Neuroscience* [Preprint]. Available at: <https://doi.org/10.1038/s41593-025-02071-5>.

Hasnain, M.A. *et al.* (2025) ‘Separating cognitive and motor processes in the behaving mouse’, *Nature Neuroscience*, 28(3), pp. 640–653. Available at: <https://doi.org/10.1038/s41593-024-01859-1>.

Mimica, B. *et al.* (2023) ‘Behavioral decomposition reveals rich encoding structure employed across neocortex in rats’, *Nature Communications*, 14(1). Available at: <https://doi.org/10.1038/s41467-023-39520-3>.

Reinert, S. *et al.* (2021) ‘Mouse prefrontal cortex represents learned rules for categorization’, *Nature*, 593(7859), pp. 411–417. Available at: <https://doi.org/10.1038/s41586-021-03452-z>.

(3) At what point were the behavioral and recording data acquired, specifically in relation to the stability of the environment? Since the environment is dynamically adjusted during training to shape behavior (as described in the Methods), it is important to clarify how long the environment had been stable at the time of data collection. While the inclusion criteria for sessions are described on pages 16–17, this specific information appears to be missing. Ideally, it should be stated clearly in the main text as the authors emphasize a non-volatile environment during data acquisition.

We thank the reviewer for raising this point. Indeed, we modified subtle task features to facilitate the animals’ learning progress. These features included, for example, the wait time in the central port that was gradually increased over the course of learning or the number of different stimulus difficulties that were introduced once animals were already experts. We thus agree with the reviewer that these aspects of the task were not stable. However, the stimulus-response contingency never changed. This means that for the entire training mice were only rewarded on the right side after experiencing a high-rate stimulus train and on the left after a low-rate stimulus train. This feature is very different from set-shifting and reversal learning tasks where the underlying decision-rules dynamically change during individual sessions. Whenever we refer to the stability of the environment in the text, we intend to highlight that the stimulus contingencies remained the same throughout training and expert performance. We have changed the main text to explicitly reflect that the rewarded contingencies remained stable throughout the entire experiment (Lines 350, 530 - 531, 578 - 580):

“We find that ACC neurons in expert mice performing a fully deterministic perceptual task without rule-switches encode trial history more strongly than other task variables.”

“Importantly, the association between stimulus rate and rewarded port remained the same throughout the entire experiment and across all sessions.”

“These animals were exposed to punishments after showing the first signs of learning at 24 ± 4 sessions on average. The absence of punishment after incorrect choices,…”

Minor comments:

Figure 2J seemed not to be referred to in the main results.

We thank the reviewer for catching this. We have added a reference to Figure 2j (Line 200), at the location in the main text where we discuss this panel.

Reviewer #5 (Remarks to the Author):

We thank the reviewer for the review and the feedback.

contingencies remained the same throughout training and expert performance. We have changed the main text to explicitly reflect that the rewarded contingencies remained stable throughout the entire experiment (Lines 350, 530 - 531, 578 - 580):

“We find that ACC neurons in expert mice performing a fully deterministic perceptual task without rule-switches encode trial history more strongly than other task variables.”

“Importantly, the association between stimulus rate and rewarded port remained the same throughout the entire experiment and across all sessions.”

“These animals were exposed to punishments after showing the first signs of learning at 24 ± 4 sessions on average. The absence of punishment after incorrect choices,…”

Minor comments:

Figure 2J seemed not to be referred to in the main results.

We thank the reviewer for catching this. We have added a reference to Figure 2j (Line 200), at the location in the main text where we discuss this panel.

Reviewer #5 (Remarks to the Author):

We thank the reviewer for the review and the feedback.

REVIEWER COMMENTS

Reviewer #1 (Remarks to the Author):

The authors have fully addressed all of my previous concerns.

We thank the reviewer for helping to strengthen our manuscript.

Reviewer #3 (Remarks to the Author):

The authors have answered many of the prior questions and supplied enough information to better understand the approach used to analyse the neuronal and behavioral data. I have no other substantial remarks, but I would like to urge that we make an effort to remove the ambiguity that remains in the definition of trial. In general, it is apparent that a "current trial" and a "prior trial" definition exist, although this is not always the case, and there is still some uncertainty (e.g., lines 148; 170; 195-196) of what a trial is. I would also propose revising the notion of "self-initiated perceptual decision making". I understand the authors' explanation, in response to my previous comment, but the phrase in this case implies that the decision is self-initiated, which is incorrect.

To clarify the message, I'd want to point up one final source of ambiguity. The definition of ITI has been complicated in the most recent version of the manuscript, used for Supplement Figure 2d. This refers to the classic ITI that separate one trial from another. This is however considerably different from the terms ITI used in "early ITI" and "late ITI" elsewhere.

We thank the reviewer for pointing out these remaining sources of possible confusion. We have revised the manuscript text and clarified the sentences at the locations pointed out by the reviewer (lines 148, 170, and 195 - 196). They now read as follows:

“For example, the activity of some neurons was driven almost exclusively by previous correct right or correct left choices during the inter-trial interval (ITI, Figure 2d, N1, N4) or by both previous incorrect left and right choices but at different times within the ITI (Figure 2d, N7), indicating clear non-linear mixed selectivity^{35,36}.”

“To answer this question, we separately computed the decoding accuracy for each trial history context across the different trial phases (Figure 2g).”

“We therefore asked whether the neural representations encoding trial history information remained stable⁴¹⁻⁴³ or whether they evolve gradually⁴⁴ across different trial phases.”

With respect to the reviewer’s remark about self-initiated trials (also Reviewer #3, minor comment #3), we agree with the reviewer that the trials in our task are not free choice trials where animals can make their decisions as soon as they deem the amount of integrated evidence sufficient. Instead, we impose a wait time to ensure that the animals are sampling the entire 1 second stimulus train at least once. However, we would like to stress that trials are self-initiated. This is fundamentally different from many other current rodent decision-making tasks (particularly under head-restraint) where trials are often automatically triggered, and subjects can engage with the task or not. To avoid any source of confusion we have removed the expression “self-initiated” from lines 79 and 390.

Finally, we have added more text to explain what entails the ITI duration on lines 204 – 205 and in the legend to supplementary figure 2:

“To test for the influence of time on trial history decoding we separated the trials based on how much time elapsed between the beginning of the early ITI and the initiation of the current trial (ITI duration)...”

“The ITI duration is defined here as the time between outcome delivery in the preceding trial and current trial initiation (see also Figure 1b).”

Reviewer #4 (Remarks to the Author):

The authors addressed all my concerns.

We thank the reviewer for the important input.

Reviewer #5 (Remarks to the Author):
